# Sym-NCO: Leveraging Symmetricity for Neural Combinatorial Optimization

**Minsu Kim**    **Junyoung Park**    **Jinkyoo Park**
Korea Advanced Institute of Science and Technology (KAIST)
Dept. Industrial & Systems Engineering
{min-su, Junyoungpark, jinkyoo.park}@kaist.ac.kr

## Abstract

Deep reinforcement learning (DRL)-based combinatorial optimization (CO) methods (i.e., DRL-NCO) have shown significant merit over the conventional CO solvers as DRL-NCO is capable of learning CO solvers less relying on problem-specific expert domain knowledge (heuristic method) and supervised labeled data (supervised learning method). This paper presents a novel training scheme, Sym-NCO, which is a regularizer-based training scheme that leverages universal symmetricities in various CO problems and solutions. Leveraging symmetricities such as rotational and reflectional invariance can greatly improve the generalization capability of DRL-NCO because it allows the learned solver to exploit the commonly shared symmetricities in the same CO problem class. Our experimental results verify that our Sym-NCO greatly improves the performance of DRL-NCO methods in four CO tasks, including the traveling salesman problem (TSP), capacitated vehicle routing problem (CVRP), prize collecting TSP (PCTSP), and orienteering problem (OP), without utilizing problem-specific expert domain knowledge. Remarkably, Sym-NCO outperformed not only the existing DRL-NCO methods but also a competitive conventional solver, the iterative local search (ILS), in PCTSP at $240\times$ faster speed. Our source code is available at https://github.com/alstn12088/Sym-NCO.

## 1 Introduction

Combinatorial optimization problems (COPs), mathematical optimization problems on discrete input space, have been used to solve numerous valuable applications, including vehicle routing problems (VRPs) [1, 2], drug discovery [3, 4], and semi-conductor design [5, 6, 7, 8, 9]. However, finding an optimal solution to COP is difficult due to its NP-hardness. Therefore, computing near-optimal solutions fast is essential from a practical point of view.

Conventionally, COPs are solved by integer program (IP) solvers or hand-crafted (meta) heuristics. Recent advances in computing infrastructures and deep learning have conceived the field of neural combinatorial optimization (NCO), a deep learning-based COP solving strategy. Depending on the training scheme, NCO methods are generally classified into supervised learning [10, 11, 12, 13, 14] and reinforcement learning (RL) [15, 16, 17, 18, 19, 20, 21, 22, 23, 24, 25, 26, 27]. Depending on the solution generation scheme, NCO methods are also classified into improvement [17, 16, 15, 28, 18, 19, 25] and constructive heuristics [20, 21, 22, 23, 24, 26, 27]. Among the NCO approaches, deep RL (DRL)-based constructive heuristics (i.e., DRL-NCO) are favored over conventional approaches for two major reasons. First, RL can be applied to train the NCO model in less explorered CO tasks because training RL does not require domain expert knowledge and supervised labels from a verified solver. Second, it is easy to produce qualified feasible solutions because the constructive process can easily avoid constraint-violated actions [21]. Despite the strength of DRL-NCO, there exists a performance gap between the state-of-the-art

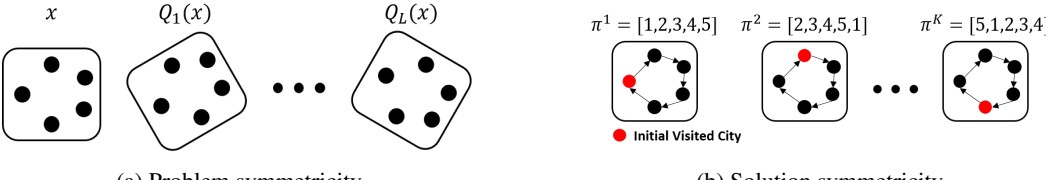

| (a) Problem symmetricity | (b) Solution symmetricity |

Figure 2: Illustration of symmetricities in CO (exampled in TSP)

conventional heuristics and DRL-NCO. In an effort to close the gap, there have been attempts to employ problem-specific heuristics to existing DRL-NCO methods [23, 29]. However, devising a general training scheme to improve the performance of DRL-NCO still remains challenging.

In this study, we propose the Symmetric Neural Combinatorial Optimization (Sym-NCO), a general training scheme applicable to universal CO problems. Sym-NCO is a regularization-based training scheme that leverages the symmetricities commonly found in COPs to increase the performance of existing DRL-NCO methods. Sym-NCO leverages two types of symmetricities innate in COP that are defined on the Euclidean graph. First, the problem symmetricity is derived from the rotational invariance of the solution; the rotated graph must exhibit the same optimal solution as the original graph as shown in Fig. 2a. Second, the solution symmetricity refers to the property that solutions have identical output values (See Fig. 2b).

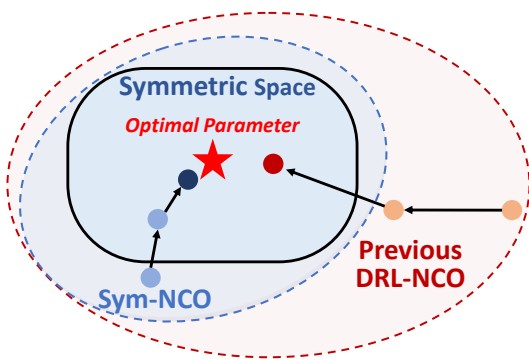

Figure 1: Illustration that describes an advantage of Sym-NCO. An optimal training parameter is in symmetric space. Sym-NCO makes a more compact training space compared with previous DRL-NCO and supports the NCO model efficiently converges in near-optimal parameters.

To train an effective NCO solver while leveraging the symmetricities, we employ REINFORCE algorithm with the baselines terms specially designed to impart solution and problem symmetricities. REINFORCE algorithm is used because the well-known effective NCO solvers are trained by REINFORCE; thus, we can improve such solvers by just modifying their baseline with our symmetricity-considered baseline terms. Specifically, we sample multiple solutions from the transformed problems and use the average return of them. Then, REINFORCE pushes each solution sampled by the solver to excel this baseline during training, thus improving the policy and making all the solutions the same, i.e., the problem and solution symmetricities are realized.

**Motivation for learning symmetricity.** Leveraging symmetricity is important to train CO models for two major reasons. Firstly, symmetricity is a strong inductive bias that can support the training process of DRL by making compact training space as shown in Fig. 1. Secondly, learning symmetricity is beneficial to increasing generalization capability for unseen CO problems because symmetricity induces the invariant representation that every COP contains.

**Novelty.** The major novelty of the proposed learning strategy is that it can easily improve existing powerful CO models; existing equivariant neural network schemes must be re-designed at the architecture level for adapting to CO.

## 2 Symmetricity in Combinatorial Optimization Markov Decision Process

This section presents several symmetric characteristics found in combinatorial optimization, which is formulated in the Markov decision process. The objective of NCO is to train the $\theta$-parameterized solver $F_\theta$ by solving the following problem:

$$\theta^* = \arg\max_\theta \mathbb{E}_{\boldsymbol{P}\sim\rho}\big[\mathbb{E}_{\boldsymbol{\pi}\sim F_\theta(\boldsymbol{P})}\big[R(\boldsymbol{\pi};\boldsymbol{P})\big]\big] \tag{1}$$

where $\boldsymbol{P} = (\boldsymbol{x}, \boldsymbol{f})$ is a problem instance with the $N$ node coordinates $\boldsymbol{x} = \{x_i\}_{i=1}^N$ and corresponding $N$ features $\boldsymbol{f} = \{f_i\}_{i=1}^N$. The $\rho$ is a problem generating distribution. The $\boldsymbol{\pi} = \{\pi_i\}_{i=1}^N$ is a solution

where each element is index value $\pi_i \in \{1, ..., N\}$ for coordinates $\boldsymbol{x} = \{x_i\}_{i=1}^N$. The $R(\boldsymbol{\pi}; \boldsymbol{P})$ is objective value for $\boldsymbol{\pi}$ on problem $\boldsymbol{P}$.

For example, assume that we solve TSP with five cities (i.e. $N = 5$). Then the problem instance $\boldsymbol{P}$ contains five city coordinates $\boldsymbol{x} = \{x_i\}_{i=1}^5$ where the salesman must visit. The solution $\boldsymbol{\pi}$ is a sequence of city indices; if $\boldsymbol{\pi} = \{1, 3, 2, 5, 4\}$, the salesman visits city coordinates as $x_1 \rightarrow x_3 \rightarrow x_2 \rightarrow x_5 \rightarrow x_4 \rightarrow x_1$ (the salesman must go back to first visited city to complete a tour). In TSP, each city contains the homogeneous features $\boldsymbol{f}$. Thus, objective $R$ of the TSP is defined as the negative of tour length: $R(\boldsymbol{\pi}; \boldsymbol{P}) = -\left(\sum_{i=1}^4 ||x_{\pi_{i+1}} - x_{\pi_i}|| + ||x_{\pi_5} - x_{\pi_1}||\right)$. This combinatorial decision process can be expressed as Markov decision process, and the solver $F_\theta(\boldsymbol{\pi}|\boldsymbol{P})$ can be expressed as *instance conditioned policy*. To this end, we can utilize deep reinforcement learning for training solver $F_\theta(\boldsymbol{\pi}|\boldsymbol{P})$. We formally define the Markov decision process for CO in the below chapter.

## 2.1 Combinatorial optimization Markov decision process

We define the combinatorial optimization Markov decision process (CO-MDP) as the sequential construction of a solution of COP. For a given $\boldsymbol{P}$, the components of the corresponding CO-MDP are defined as follows:

- **State.** The state $\boldsymbol{s}_t = (\boldsymbol{a}_{1:t}, \boldsymbol{x}, \boldsymbol{f})$ is the $t$-th (partially complete) solution, where $\boldsymbol{a}_{1:t}$ represents the previously selected nodes. The initial and terminal states $\boldsymbol{s}_0$ and $\boldsymbol{s}_T$ are equivalent to the empty and completed solution, respectively. In this paper, we denote the solution $\boldsymbol{\pi}(\boldsymbol{P})$ as the completed solution.

- **Action.** The action $a_t$ is the selection of a node from the un-visited nodes (i.e., $a_t \in \mathbb{A}_t = \{\{1, ..., N\} \setminus \{\boldsymbol{a}_{1:t-1}\}\}$).

- **Reward.** The reward function $R(\boldsymbol{\pi}; \boldsymbol{P})$ maps the objective value from given $\boldsymbol{\pi}$ of problem $\boldsymbol{P}$. We assume that the reward is a function of $\boldsymbol{a}_{1:T}$ (solution sequences), $||x_i - x_j||_{i,j \in \{1,...N\}}$ (relative distances) and $\boldsymbol{f}$ (nodes features). In TSP, capacitated VRP (CVRP), and prize collecting TSPs (PCTSP), the reward is the negative of the tour length. In orienteering problem (OP), the reward is the sum of the prizes.

Having defined CO-MDP, we define the solution mapping as follows: $\boldsymbol{\pi} \sim F_\theta(\cdot|P) = \prod_{t=1}^T p_\theta(a_t|\boldsymbol{s}_t)$ where $p_\theta(a_t|\boldsymbol{s}_t)$ is the policy that produces $a_t$ at $\boldsymbol{s}_t$, and $T$ is the maximum number of states in the solution construction process.

## 2.2 Symmetricities in CO-MDP

Symmetricities are found in various COPs. We conjecture that imposing those symmetricities on $F_\theta$ improves the generalization and sample efficiency of $F_\theta$. We define the two identified symmetricities that are commonly found in various COPs:

**Definition 2.1 (Problem Symmetricity).** Problem $\boldsymbol{P}^i$ and $\boldsymbol{P}^j$ are problem symmetric ($\boldsymbol{P}^i \overset{\text{sym}}{\longleftrightarrow} \boldsymbol{P}^j$) if their optimal solution sets are identical.

**Definition 2.2 (Solution Symmetricity).** Two solutions $\boldsymbol{\pi}^i$ and $\boldsymbol{\pi}^j$ are solution symmetric ($\boldsymbol{\pi}^i \overset{\text{sym}}{\longleftrightarrow} \boldsymbol{\pi}^j$) on problem $\boldsymbol{P}$ if $R(\boldsymbol{\pi}^i; \boldsymbol{P}) = R(\boldsymbol{\pi}^j; \boldsymbol{P})$.

An exemplary problem symmetricity found in various COPs is the rotational symmetricity:

**Theorem 2.1 (Rotational symmetricity).** For any orthogoanl matrix $Q$, the problem $\boldsymbol{P}$ and $Q(\boldsymbol{P}) \triangleq \{\{Qx_i\}_{i=1}^N, \boldsymbol{f}\}$ are problem symmetric: i.e., $\boldsymbol{P} \overset{\text{sym}}{\longleftrightarrow} Q(\boldsymbol{P})$. See Appendix A for the proof.

Rotational problem symmetricity is identified in every Euclidean COPs. On the other hand, solution symmetricity cannot be identified easily as the properties of the solutions are distinct for every COP.

# 3 Symmetric Neural Combinatorial Optimization

This section presents Sym-NCO, an effective training scheme that leverages the symmetricities of COPs. Sym-NCO learns a solve $F_\theta$ by minimizing the total loss function:

$$\mathcal{L}_{\text{total}} = \mathcal{L}_{\text{Sym-RL}} + \alpha\mathcal{L}_{\text{inv}} = \mathcal{L}_{\text{ps}} + \beta\mathcal{L}_{\text{ss}} + \alpha\mathcal{L}_{\text{inv}} \tag{2}$$

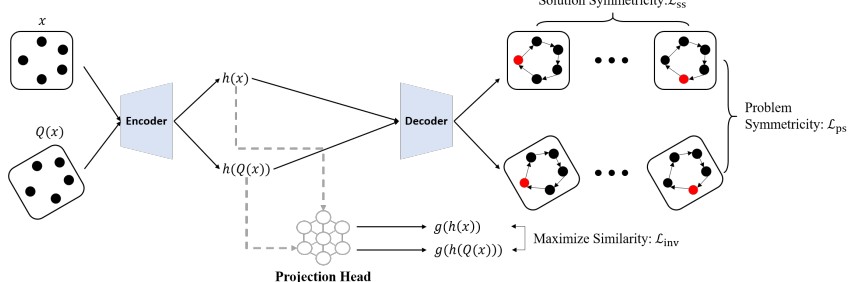

Figure 3: An overview of Sym-NCO

where $\mathcal{L}_{\text{Sym-RL}}$ is REINFORCE loss term supported by symmetricity and $\mathcal{L}_{\text{inv}}$ is the regularization loss to induce that invariant representation. The $\mathcal{L}_{\text{Sym-RL}}$ is composed with $\mathcal{L}_{\text{ss}}$, the REINFORCE loss term of Eq. (1) with solution symmetricity regularizing mechanism, and $\mathcal{L}_{\text{ps}}$, the REINFORCE loss term of Eq. (1) with both solution and problem symmetricity regularization. $\alpha, \beta \in [0, 1]$ are the weight coefficients. In the following subsections, we explain each loss term in detail.

## 3.1 Regularizing REINFORCE with problem and solution symmetricities via $\mathcal{L}_{\text{Sym-RL}}$

As discussed in Section 2.2, COPs have problem and solution symmetricities. We explain how to learn solver $F_\theta$ using REINFORCE with the specially designed baseline to approximately impose symmetricities. We provide the policy gradients to $\mathcal{L}_{\text{ss}}(\pi(\boldsymbol{P}))$ and $\mathcal{L}_{\text{ps}}(\pi(\boldsymbol{P}))$ in the context of the REINFORCE algorithm [30] with the proposed baseline schemes.

**Leveraging solution symmetricity.** As defined in Definition 2.2, the symmetric solutions must have the same objective values. We propose the REINFORCE loss $\mathcal{L}_{\text{ss}}$ with the baseline specially designed to exploit the solution symmetricity of CO as follows:

$$\mathcal{L}_{\text{ss}} = -\mathbb{E}_{\boldsymbol{\pi} \sim F_\theta(\cdot|\boldsymbol{P})}\big[R(\boldsymbol{\pi}; \boldsymbol{P})\big] \tag{3}$$

$$\nabla_\theta \mathcal{L}_{\text{ss}} = -\mathbb{E}_{\boldsymbol{\pi} \sim F_\theta(\cdot|\boldsymbol{P})}\Big[\big[R(\boldsymbol{\pi}; \boldsymbol{P}) - b(\boldsymbol{P})\big]\nabla_\theta \log F_\theta\Big] \tag{4}$$

$$\approx -\frac{1}{K}\sum_{k=1}^{K}\Big[\big[R(\boldsymbol{\pi}^k; \boldsymbol{P}) - \frac{1}{K}\sum_{k=1}^{K}R(\boldsymbol{\pi}^k; \boldsymbol{P})\big]\nabla_\theta \log F_\theta\Big] \tag{5}$$

where $\{\boldsymbol{\pi}^k\}_{k=1}^K$ are the solutions of $P$ sampled from $F_\theta(\boldsymbol{\pi}|\boldsymbol{P})$, $\log F_\theta$ is the log-likelihood of $F_\theta$, $K$ is the number of sampled solutions, $b(\boldsymbol{P})$ is a shared baseline which is the average reward from $K$ solutions for the identical problem $\boldsymbol{P}$.

The $\mathcal{L}_{\text{ss}}$ approximately imposes solution symmetricity using REINFORCE algorithm with a novel baseline $b(\boldsymbol{P})$. The sum of advantage in the solution group $\{\boldsymbol{\pi}^k\}_{k=1}^K$ is always zero:

$$\frac{1}{K}\sum_{k=1}^{K}\Big[\big[R(\boldsymbol{\pi}^k; \boldsymbol{P}) - \frac{1}{K}\sum_{k=1}^{K}R(\boldsymbol{\pi}^k; \boldsymbol{P})\big]\Big] = 0 \tag{6}$$

The $\mathcal{L}_{\text{ss}}$ induces competition among the rewards, $R(\boldsymbol{\pi}^1; \boldsymbol{P}), ..., R(\boldsymbol{\pi}^K; \boldsymbol{P})$ which can be seen as a zero-sum game. Therefore, $\mathcal{L}_{\text{ss}}$ improves the overall reward quality of the solution group using the proposed competitive REINFORCE scheme, making the solver generate high-rewarded solutions but a small reward deviation between solutions. The small reward-deviation indicates $\mathcal{L}_{\text{ss}}$ approximately imposes solution symmetricity to solver $F_\theta$.

The POMO [23] employed a similar training technique with our $\mathcal{L}_{\text{ss}}$, that finds symmetric solutions by forcing $F_\theta$ to visit all possible initial cities when solving TSP and CVRP. However, the reward of COPs, including CVRP, PCTSP, and OP, is usually sensitive to first-city selection. Therefore, POMO can be an excellent standalone method for TSP but can be further improved using our $\mathcal{L}_{\text{ss}}$ loss term in other tasks including CVRP.

**Leveraging problem symmetricity.** As discussed in Section 2.2, the rotational problem symmetricity is common in various COPs. We propose the REINFORCE loss $\mathcal{L}_{\text{ps}}$ which is equipped with problem

symmetricity:

$$\mathcal{L}_{\text{ps}} = -\mathbb{E}_{Q^l \sim \mathbf{Q}} \mathbb{E}_{\boldsymbol{\pi} \sim F_\theta(\cdot | Q^l(\boldsymbol{P}))} \big[ R(\boldsymbol{\pi}; \boldsymbol{P}) \big] \tag{7}$$

$$\nabla_\theta \mathcal{L}_{\text{ps}} = -\mathbb{E}_{Q^l \sim \mathbf{Q}} \bigg[ \mathbb{E}_{\boldsymbol{\pi} \sim F_\theta(\cdot | Q^l(\boldsymbol{P}))} \big[ \big[ R(\boldsymbol{\pi}; \boldsymbol{P}) - b(\boldsymbol{P}, \boldsymbol{Q}) \big] \nabla_\theta \log F_\theta \big] \bigg] \tag{8}$$

$$\approx \frac{1}{LK} \sum_{l=1}^{L} \sum_{k=1}^{K} \bigg[ \big[ R(\boldsymbol{\pi}^{l,k}; \boldsymbol{P}) - \frac{1}{LK} \sum_{l=1}^{L} \sum_{k=1}^{K} R(\boldsymbol{\pi}^{l,k}; \boldsymbol{P}) \big] \nabla_\theta \log F_\theta \bigg] \tag{9}$$

where $\mathbf{Q}$ is the distribution of random orthogonal matrices, $Q^l$ is the $l^{\text{th}}$ sampled rotation matrix, and $\boldsymbol{\pi}^{l,k}$ is the $k^{\text{th}}$ sample solution of the $l^{\text{th}}$ rotated problem. We construct $L$ problem symmetric problems, $Q^1(\boldsymbol{P}), ..., Q^L(\boldsymbol{P})$, by using the sampled rotation matrices, and smaple $K$ symmetric solutions from each of the $L$ problems. Then, the shared baseline $b(\boldsymbol{P}, \boldsymbol{Q})$ is constructed by averaging $L \times K$ solutions.

Similar to the regularization scheme of $\mathcal{L}_{\text{ss}}$, the advantage term of $\mathcal{L}_{\text{ps}}$ also induces competition between solutions sampled from rotationally symmetric problems. Since the rotational symmetricity is defined such that $x$ and $Q_l(x)$ have the same solution, the negative advantage value forces the solver to find a better solution. As mentioned in Section 2.2, problem symmetricity in COPs is usually pre-identified (i.e. there is provable guaranteed symmetricity such as rotational symmetricity Theorem 2.1); $\mathcal{L}_{\text{ps}}$ are applicable to general COPs. Moreover, multiple solutions are sampled for each symmetric problem so that $\mathcal{L}_{\text{ps}}$ can also identify and exploit the solution symmetricity with a similar approach taken for $\mathcal{L}_{\text{ss}}$. We provide detailed implementation and design guides regarding the integration strategy of $\mathcal{L}_{\text{ss}}$ and $\mathcal{L}_{\text{ps}}$ in Appendix C.2.

### 3.2 Learning invariant representation with Pre-identified Symmetricity: $\mathcal{L}_{\text{inv}}$.

By Theorem 2.1, the original problem $x$ and its rotated problem $Q(x)$ have identical solutions. Therefore the encoder of $F_\theta$ can be enforced to have invariant representation by leveraging the pre-identified symmetricity: rotation symmetricity.

We denote $h(x)$ and $h(Q(x))$ as the hidden representations of $x$ and $Q(x)$, respectively. To impose the rotational invariant property on $h(x)$, we train solver $F_\theta$ with an additional regularization loss term $\mathcal{L}_{\text{inv}}$ defined as:

$$\mathcal{L}_{\text{inv}} = -S_{\cos}\Big( g\big( h(x) \big), g\big( h(Q(x)) \big) \Big) \tag{10}$$

where $S_{\cos}(a, b)$ is the cosine similarity between $a$ and $b$. $g$ is the MLP-parameterized projection head.

For learning the invariant representation on rotational symmetricity, we penalize the difference between the projected representation $g(h(x))$ and $g(h(Q(x)))$, instead of directly penalizing the difference between $h(x)$ and $h(Q(x))$. This penalizing scheme allows the use of an arbitrary encoder network architecture while maintaining the diversity of $h$ [31]. We empirically verified that this approach attains stronger solvers as described in Section 6.1.

## 4 Related Works

**Deep construction heuristics.** Bello et al. [20] propose one of the earliest DRL-NCO methods, based on pointer network (PointerNet) [10], and trained it with an actor-critic method. Attention model (AM) [21] successfully extends [20] by swapping PointerNet with Transformer [32], and it is currently the *de-facto* standard method for NCO. Notably, AM verifies its problem agnosticism by solving several classical routing problems and their practical extensions [7, 19]. The multi-decoder AM (MDAM) [26] extends AM by employing an ensemble of decoders. However, such an extension is inapplicable for stochastic routing problems. The policy optimization for multiple optimal (POMO) [23] extends AM by exploiting the solution symmetricities in TSP and CVRP. Even though POMO shows significant improvements from AM, it relies on problem-specific solution symmetricities for TSP. Our method can be seen as a general-purpose symmetric learning scheme, extended from POMO, which can be applied in more general CO tasks.

**Equivariant deep learning.** In deep learning, symmetricities are often enforced by employing specific network architectures. Niu et al. [33] proposes a permutation equivariant graph neural network (GNN) that produces equivariant outputs to the input order permutations. The $SE(3)$-Transformer [34] restricts the Transformer so that it is equivariant to $SE(3)$ group input transformation. Similarly, equivariant GNN (EGNN) [35] proposes a GNN architecture that produces $O(n)$ group equivariant output. These network architectures can dramatically reduce the search space of the model parameters. Some research applies equivariant neural networks to RL tasks to improve sample efficiency [36]. Also, there are several works that exploited the symmetric nature of CO. Ouyang et al. [37] proposed equivariant encoding scheme by giving rule-based input transformation to input graph. Hudson et al. [38] suggested a line-graph embedding scheme, which is beneficial to process CO graphs with rotational equivariant. Our Sym-NCO is a regularization method that is capable of learning existing powerful CO models without giving rule-based hard constraints to the structure. We empirically study the benefit of Sym-NCO over other symmetricity-based approaches in Section 6.1 and Appendix D.5.

## 5 Experiments

This section provides the experimental results of Sym-NCO for TSP, CVRP, PCTSP, and OP. Focusing on the fact that Sym-NCO can be applied to any encoder-decoder-based NCO method, we implement Sym-NCO on top of POMO [23] to solve TSP and CVRP, and AM [21] to solve PCTSP and OP, respectively. We additionally validate the effectiveness of Sym-NCO on PointerNet [10] at TSP ($N = 100$).

### 5.1 Tasks and baseline selections

TSP aims to find the Hamiltonian cycle with a minimum tour length. We employ Concorde [39] and LKH-3 [40] as the non-learnable baselines, and PointerNet [10], the structured-to-vector deep-Q-network (S2V-DQN) [41], AM [21], POMO [23] and MDAM [26] as the neural constructive baselines.

CVRP is an extension of TSP that aims to find a set of tours with minimal total tour lengths while satisfying the capacity limits of the vehicles. We employ LKH-3 [40] as the non-learnable baselines, and Nazari et al. [22], AM [21], POMO [23] and MDAM [26] as the constructive neural baselines.

PCTSP is a variant of TSP that aims to find a tour with minimal tour length while satisfying the prize constraints. We employ the iterative local search (ILS) [21] as the non-learnable baseline, and AM [21] and MDAM [26] as the constructive neural baselines.

OP is a variant of TSP that aims to find the tour with maximal total prizes while satisfying the tour length constraint. We employ *compass* [42] as the non-learnable baseline, and AM [21] and MDAM [26] as the constructive neural baselines.

### 5.2 Experimental setting

**Problem size.** We provide the results of problems with $N = 100$ for the four problem classes, and real-world TSP problems with $50 < N < 250$ from TSPLIB.

**Hyperparameters.** We apply Sym-NCO to POMO, AM, and PointerNet. To make fair comparisons, we use the same network architectures and training-related hyperparameters from their original papers to train their Sym-NCO-augmented models. Please refer to Appendix Appendix C.1 for more details.

**Dataset and Computing Resources.** We use the benchmark dataset [21] to evaluate the performance of the solvers. To train the neural solvers, we use *Nvidia* A100 GPU. To evaluate the inference speed, we use an *Intel* Xeon E5-2630 CPU and *Nvidia* RTX2080Ti GPU to make fair comparisons with the existing methods as proposed in [26].

### 5.3 Performance metrics

This section provides detailed performance metrics:

**Average cost.** We report an average cost of 10,000 benchmark instances which is proposed by [21].

Table 1: Performance evaluation results for TSP and CVRP. Bold represents the best performances in each task. '-' indicates that the solver does not support the problem. 's' indicates multi-start sampling, 'bs' indicates the beam search. '×5 for the MDAM indicates the 5 decoder ensemble.

| Method | | TSP ($N = 100$) | | | CVRP ($N = 100$) | | |
|---|---|---|---|---|---|---|---|
| | | Cost ↓ | Gap | Time | Cost ↓ | Gap | Time |
| *Handcrafted Heuristic-based Classical Methods* | | | | | | | |
| Concorde | Heuristic [39] | 7.76 | 0.00% | 3m | | – | |
| LKH3 | Heuristic [40] | 7.76 | 0.00% | 21m | 15.65 | 0.00% | 13h |
| *RL-based Deep Constructive Heuristic methods with greedy rollout* | | | | | | | |
| PointerNet {*greedy.*} | NIPS'15 [10, 20] | 8.60 | 6.90 % | – | | – | |
| S2V-DQN {*greedy.*} | NIPS'17 [41] | 8.31 | 7.03 % | – | | | |
| RL {*greedy.*} | NeurIPS'18 [22] | | – | | 17.23 | 10.12% | – |
| AM {*greedy.*} | ICLR'19 [21] | 8.12 | 4.53% | 2s | 16.80 | 7.34% | 3s |
| MDAM {*greedy.*× 5} | AAAI'21 [25] | 7.93 | 2.19% | 36s | 16.40 | 4.86% | 45s |
| POMO {*greedy.*} | NeurIPS'20 [23] | 7.85 | 1.04% | 2s | 16.26 | 3.93% | 3s |
| **Sym-NCO** {*greedy.*} | *This work* | **7.84** | **0.94**% | 2s | **16.10** | **2.88**% | 3s |
| *RL-based Deep Constructive Heuristic methods with multi-start rollout* | | | | | | | |
| Nazari et al. {*bs.*10} | NeurIPS'18 [22] | | – | | 16.96 | 8.39% | – |
| AM {*s.*1280} | ICLR'19 [21] | 7.94 | 2.26% | 41m | 16.23 | 3.72% | 54m |
| POMO {s. 100} | NeurIPS'20 [23] | 7.80 | 0.44% | 13s | 15.90 | 1.67% | 16s |
| MDAM {bs. 30 × 5} | AAAI'21 [25] | 7.80 | 0.48% | 20m | 16.03 | 2.49% | 1h |
| **Sym-NCO** {*s.*100} | *This work* | **7.79** | **0.39**% | 13s | **15.87** | **1.46%** | 16s |

Table 2: Performance evaluation results for PCTSP and OP. Notations are the same with Table 1.

| Method | | PCTSP ($N = 100$) | | | OP ($N = 100$) | | |
|---|---|---|---|---|---|---|---|
| | | Cost ↓ | Gap | Time | Obj ↑ | Gap | Time |
| *Handcrafted Heuristic-based Classical Methods* | | | | | | | |
| ILS C++ | Heuristic [21] | 5.98 | 0.00% | 12h | | – | |
| Compass | Heuristic [42] | | – | | 33.19 | 0.00% | 15m |
| *RL-based Deep Constructive Heuristic methods with greedy rollout (zero-shot inference)* | | | | | | | |
| AM {*greedy.*} | ICLR'19 [21] | 6.25 | 4.46% | 2s | 31.62 | 4.75% | 2s |
| MDAM {*greedy.*× 5} | AAAI'21 [25] | 6.17 | 3.13% | 34s | 32.32 | 2.61% | 32s |
| **Sym-NCO** {*greedy.*} | *This work* | **6.05** | **1.23%** | 2s | **32.51** | **2.03%** | 2s |
| *RL-based Deep Constructive Heuristic methods with multi-start rollout (Post-processing)* | | | | | | | |
| AM {s. 1280} | ICLR'19 [21] | 6.08 | 1.67% | 27m | 32.68 | 1.55% | 25m |
| MDAM {bs. 30× 5} | AAAI'21 [25] | 6.07 | 1.46% | 16m | 32.91 | 0.84% | 14m |
| **Sym-NCO** {s. 200} | *This work* | **5.98** | **-0.02%** | 3m | **33.04** | **0.45%** | 3m |

**Evaluation speed.** We report the evaluation speeds of solvers in a out-of-the-box manner as they are used in practice. In that regard, the execution time of non-neural and neural methods are measured on CPU and GPU, respectively.

**Greedy/Multi-start performance.** For neural solvers, it is a common practice to measure *multi-start* performance as its final performance. However, when those are employed in practice, such resource consuming multi-start may not be possible. Hence, we discuss greedy and multi-start separately.

## 5.4 Experimental results

**Results of TSP and CVRP.** As shown in Table 1, Sym-NCO outperforms the NCO baselines in both the greedy rollout and multi-start settings with the fastest inference speed. Remarkably, Sym-NCO achieves a $0.95\%$ gap in TSP using the greedy rollout. In the TSP greedy setting, it solves TSP 10,000 instances in a few seconds.

**Results of PCTSP and OP.** As shown in Table 2, Sym-NCO outperforms the NCO baselines in both the greedy rollout and multi-start settings. In the multi-start setting, Sym-NCO outperforms the classical PCTSP baseline (i.e., ILS) with the $\frac{43200}{180} \approx 240\times$ faster speed.

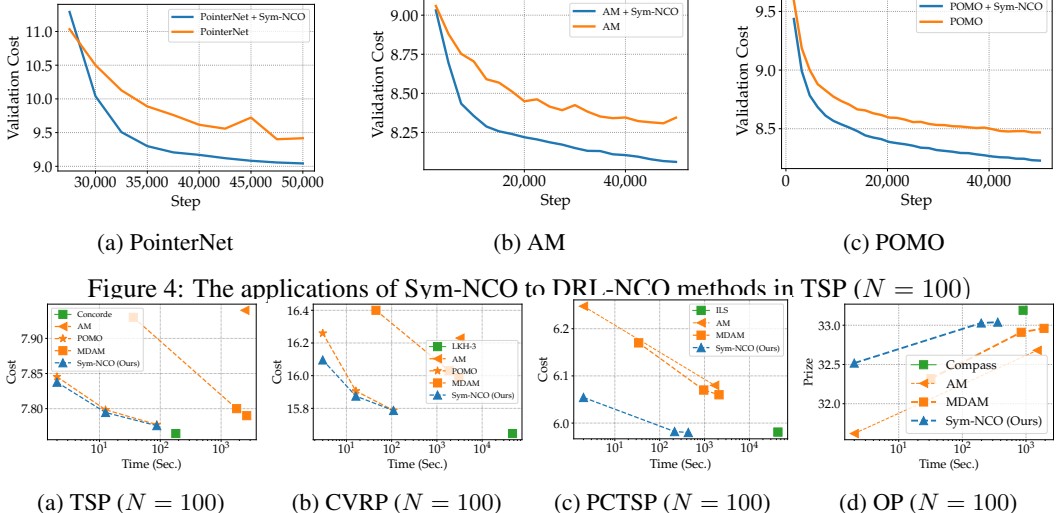

Figure 4: The applications of Sym-NCO to DRL-NCO methods in TSP ($N = 100$)

(a) TSP ($N = 100$)    (b) CVRP ($N = 100$)    (c) PCTSP ($N = 100$)    (d) OP ($N = 100$)

Figure 5: Time vs. cost plots. Green, orange, and blue colored lines visualize the results of hand-craft heuristics, neural baselines, and Sym-NCO, respectively. For OP (d), higher y-axis values are better.

**Results of the real-world TSP.** We evaluate POMO and Sym-NCO on TSPLib [43]. Table 3 shows that Sym-NCO outperforms POMO. Please refer to Appendix D.2 for the full benchmark results.

| | Gap |
|---|---|
| POMO | 1.87% |
| Sym-NCO | **1.62%** |

Table 3: Optimality gap on TSPLIB

**Application to various DRL-NCO methods.** As discussed in Section 3, Sym-NCO can be applied to various various DRL-NCO methods. We validate that Sym-NCO significantly improves the existing DRL-NCO methods as shown in Fig. 4.

**Time-performance analysis for multi-starts.** Multi-starts is a common method that improves the solution qualities while requiring a longer time budget. We use the rotation augments [23] to produce multiple inputs (i.e., starts). As shown in Fig. 5, Sym-NCO achieves the Pareto frontier for all benchmark datasets. In other words, Sym-NCO exhibits the best solution quality among the baselines within the given time consumption.

## 6 Discussion

### 6.1 Discussion of Regularization based Symmetricity Learning

**Ablation Study of $\mathcal{L}_{\mathbf{inv}}$.** As shown in Fig. 6b, $\mathcal{L}_{\text{inv}}$ increases the cosine similarity of the projected representation (i.e., $g(h)$). We can conclude that $\mathcal{L}_{inv}$ contributes to the performance improvements (see Fig. 6a). We further verify that imposing similarity on $h$ degrades the performance as demonstrated in Fig. 6c. This proves the importance of maintaining the expression power of the encoder as we mentioned in Section 3.2.

**Comparison with EGNN.** EGNN [35] provably guarantees to process coordinate graph with rotational equivalency. Also, EGNN empirically verified its' high performance in point cloud tasks. Therefore, we implemented a simple EGNN-based CO policy (simply termed EGNN in this paper), to check the feasibility of CO. We leverage six EGNN layers with 128 hidden dimensions, to replace the POMO encoder where the POMO decoder is unchanged.

In the experimental results, we observed that EGNN significantly underperforms Sym-NCO and fails to converge as shown Fig. 7b. This is because the euclidian CO has a fully connected input graph containing informative coordinates, and we believe the equivariant neural network should be carefully crafted to consider such unique input graph structures of CO tasks. On the other hand, Sym-NCO could leverage the existing powerful NCO model without fine modification of the neural network. These numerical results conform well with our hypothesis that equivariance is necessary but not

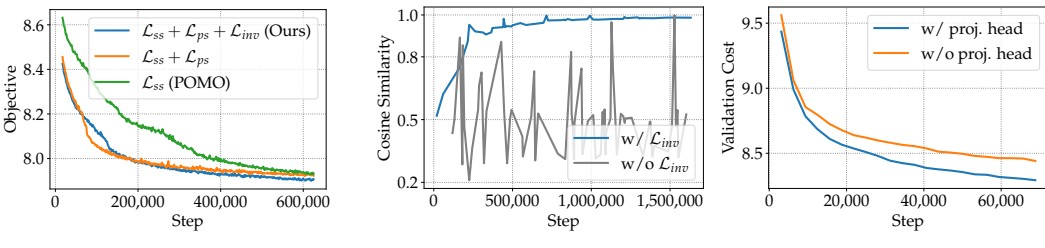

(a) Cost curves of the loss designs     (b) Cosine similarity curves     (c) Projection head ablation results

Figure 6: Loss design ablation results (a) Effect of loss components to the costs, (b) Cosine similarity curves of the models with and with $\mathcal{L}_{\text{inv}}$, (c) Costs of the models with and without $g(\cdot)$.

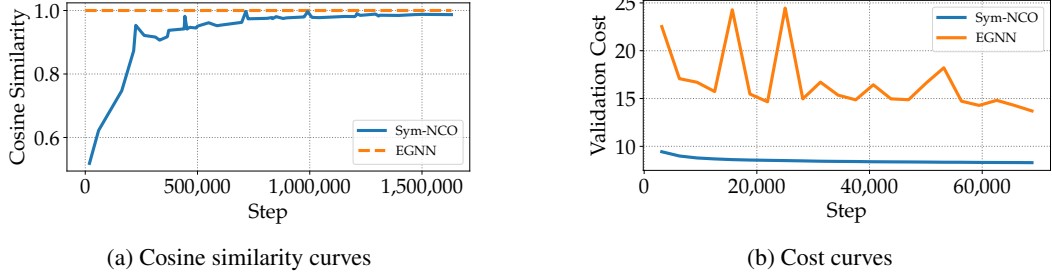

(a) Cosine similarity curves           (b) Cost curves

Figure 7: Comparisons of Sym-NCO and EGNN

sufficient for obtaining the optimal parameters for the solver. Therefore, we expect our approach can be simply extended to other domains requiring symmetricity and geometricity as positioned to leverage the existing legacy of powerful non-equivariant neural networks.

### 6.2 Limitations & future directions

**Extended problem symmetricities.** In this work, we employ the rotational symmetricity (Theorem 2.1) as the problem symmetricity. However, for some COPs, different problem symmetricities, such as scaling and translating $P$, can also be considered. Employing these additional symmetricities may further enhance the performance of Sym-NCO. We leave this for future research.

**Large scale adaptation.** Large scale applicability is essential to NCO. In this work we simply present scale adaptation capability of Sym-NCO using the effective active search (EAS) [44]; see Appendix D.3. We expect curriculum- [45], and meta-learning approaches may improve the generalizability of NCO to larger-sized problems.

**Extension to the graph COP.** This work finds the problem symmetricity that is universally applicable for *Euclidean* COPs. However, some COPs are defined in non-Euclidean spaces such as asymmetric TSP. Furthermore, there are also a bunch of existing neural combinatorial optimization models that can solve graph COP [46, 47, 48, 49, 50, 51], where we can improve with a symmetricity regularization scheme. We also leave finding the universal symmetricities of non-Euclidean COPs and applying them to existing graph COP models for future research.

### 6.3 Social Impacts

Design automation through NCO research affects various industries including logistics and transportation industries. From a negative perspective, this automation process may have some concerns to lead to unemployment in certain jobs. However, automation of logistics, transportation, and design automation can increase the efficiency of industries, reducing $CO_2$ emissions (by reducing total tour length) and creating new industries and jobs.

## Acknowledgments and Disclosure of Funding

We thank Jiwoo Son, Hyeonah Kim, Haeyeon Rachel Kim, Fangying Chen, and anonymous reviews for proving helpful feedback for preparing our manuscripts. This work was supported by a grant of the KAIST-KT joint research project through AI2XL Laboratory, Institute of convergence Technology, funded by KT [Project No. G01210696, Development of Multi-Agent Reinforcement Learning Algorithm for Efficient Operation of Complex Distributed Systems].

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
