# A Proof of Theorem 2.1

In this section, we prove the Theorem 2.1, which states a problem $\boldsymbol{P}$ and its' orthogonal transformed problem $Q(\boldsymbol{P}) = \{\{Qx_i\}_{i=1}^N, \boldsymbol{f}\}$ have identical optimal solutions if $Q$ is orthogonal matrix: $QQ^T = Q^TQ = I$.

As we mentioned in Section 2.2, reward $R$ is a function of $\boldsymbol{a}_{1:T}$ (solution sequences), $||x_i - x_j||_{i,j \in \{1,...N\}}$ (relative distances) and $\boldsymbol{f}$ (nodes features).

For simple notation, let denote $||x_i - x_j||_{i,j \in \{1,...N\}}$ as $||x_i - x_j||$. And Let $R^*(\boldsymbol{P})$ is optimal value of problem $\boldsymbol{P}$: i.e.

$$R^*(\boldsymbol{P}) = R(\boldsymbol{\pi}^*; \boldsymbol{P}) = R\left(\boldsymbol{\pi}^*; \{||x_i - x_j||, \boldsymbol{f}\}\right)$$

Where $\pi^*$ is an optimal solution of problem $\boldsymbol{P}$. Then the optimal value of transformed problem $Q(\boldsymbol{P})$, $R^*(Q(\boldsymbol{P}))$ is invariant:

$$
\begin{aligned}
R^*(Q(\boldsymbol{P})) &= R(\boldsymbol{\pi}^*; Q(\boldsymbol{P})) \\
&= R\left(\boldsymbol{\pi}^*; \{||Qx_i - Qx_j||, \boldsymbol{f}\}\right) \\
&= R\left(\boldsymbol{\pi}^*; \{\sqrt{(Qx_i - Qx_j)^T(Qx_i - Qx_j)}, \boldsymbol{f}\}\right) \\
&= R\left(\boldsymbol{\pi}^*; \{\sqrt{(x_i - x_j)^T Q^T Q(x_i - x_j)}, \boldsymbol{f}\}\right) \\
&= R\left(\boldsymbol{\pi}^*; \{\sqrt{(x_i - x_j)^T I(x_i - x_j)}, \boldsymbol{f}\}\right) \\
&= R\left(\boldsymbol{\pi}^*; \{||x_i - x_j||, \boldsymbol{f}\}\right) = R(\boldsymbol{\pi}^*; \boldsymbol{P}) = R^*(\boldsymbol{P})
\end{aligned}
$$

Therefore, problem transformation of orthogonal matrix $Q$ does not change the optimal value.

Then, the remaining proof is to show $Q(\boldsymbol{P})$ has an identical solution set with $\boldsymbol{P}$.

Let optimal solution set $\Pi^*(P) = \{\boldsymbol{\pi}^i(\boldsymbol{P})\}_{i=1}^M$, where $\boldsymbol{\pi}^i(\boldsymbol{P})$ indicates optimal solution of $\boldsymbol{P}$ and $M$ is the number of heterogeneous optimal solution.

For any $\boldsymbol{\pi}^i(Q(\boldsymbol{P})) \in \Pi^*(Q(\boldsymbol{P}))$, they have same optimal value with $\boldsymbol{P}$:

$$R(\boldsymbol{\pi}^i(Q(\boldsymbol{P})); Q(\boldsymbol{P})) = R^*(Q(\boldsymbol{P})) = R^*(\boldsymbol{P})$$

Thus, $\boldsymbol{\pi}^i(Q(\boldsymbol{P})) \in \Pi^*(P)$.

Conversely, For any $\boldsymbol{\pi}^i(\boldsymbol{P}) \in \Pi^*(\boldsymbol{P})$, they have sample optimal value with $Q(\boldsymbol{P})$:

$$R(\boldsymbol{\pi}^i(\boldsymbol{P}); \boldsymbol{P}) = R^*(\boldsymbol{P}) = R^*(Q(\boldsymbol{P}))$$

Thus, $\boldsymbol{\pi}^i(\boldsymbol{P}) \in \Pi^*(Q(\boldsymbol{P}))$.

$$\text{Therefore, } \Pi^*(\boldsymbol{P}) = \Pi^*(Q(\boldsymbol{P})), \text{ i.e., } \boldsymbol{P} \xleftrightarrow{\text{sym}} Q(\boldsymbol{P}).$$

# B  Implementation of Baselines

We directly reproduce competitive DRL-NCO methods: POMO [23] and AM [21] and PointerNet [10, 20].

**PointerNet.** The PointerNet is early work of DRL-NCO using LSTM-based encoder-decoder architecture trained with actor-critic manner. We follow the instruction of open source code [1] by [21] following hyperparmeters.

| REINFORCE baseline | Rollout baseline [21] |
|---|---|
| Learning rate | 1e-4 |
| The Number of Encoder Layer | 3 |
| Embedding Dimension | 128 |
| Batch-size | 512 |
| Epochs | 100 |
| Epoch size | 1,280,000 |
| The Number of Steps | $250K$ |

Table 4: Hyperparameter Setting for AM for all tasks.

**AM.** The AM is a general-purpose DRL-NCO, a transformer-based encoder-decoder model that solves various routing problems such as TSP, CVRP, PCTSP, and OP. We follow the instruction of open source code, same with the PointerNet with the following hyperparameters.

| REINFORCE baseline | Rollout baseline [21] |
|---|---|
| Learning rate | 1e-4 |
| The Number of Encoder Layer | 3 |
| Embedding Dimension | 128 |
| Attention Head Number | 8 |
| Feed Forward Dimension | 512 |
| Batch-size | 512 |
| Epochs | 100 |
| Epoch size | 1,280,000 |
| The Number of Steps | $250K$ |

Table 5: Hyperparameter Setting for AM for all tasks.

**POMO.** The POMO is a high-performance DRL-NCO for TSP and CVRP, implemented on the top of the AM. We follow the instruction of open source code [2] with the following hyperparameters.

| | TSP | CVRP |
|---|---|---|
| REINFORCE baseline | POMO shared baseline [23] | |
| Learning rate | 1e-4 | |
| Weight decay | 1e-6 | |
| The Number of Encoder Layer | 6 | |
| Embedding Dimension | 128 | |
| Attention Head Number | 8 | |
| Feed Forward Dimension | 512 | |
| Batch-size | 64 | |
| Epochs | 2,000 | 8,000 |
| Epoch size | 100,000 | 10,000 |
| The Number of Steps | $3.125M$ | $1.25M$ |

Table 6: Hyperparameter Setting for POMO in TSP and CVRP.

---

[1] https://github.com/wouterkool/attention-learn-to-route
[2] https://github.com/yd-kwon/POMO

# C Implementation Details of Proposed Method

## C.1 Training Hyperparameters

Sym-NCO is a training scheme that is attached to the top of the existing DRL-NCO model. We set the same hyperparameters with PointerNet, AM, and POMO Appendix B except REINFORCE baseline (we set the proposed Sym-NCO baseline introduced in Section 3).

Sym-NCO has additional hyperparameters. First of all, we set identical hyperparameters for Pointer-Net and AM for all tasks:

| | |
|---|---|
| $\alpha$ | 0.1 |
| $\beta$ | 0 |
| $K$ | 1 |
| $L$ | 10 |

Table 7: Hyperparameter Setting of Sym-NCO for PointerNet and AM.

Note that the design choice of $\beta = 0$ is to show high applicability of $\mathcal{L}_{\text{ps}}$, and is because AM with $\mathcal{L}_{\text{ss}}$ is just similar to the POMO.

For POMO, we set $\beta = 1$ to force solution symmetricity on the top of POMO's baseline. Note that we follow POMO's first node restriction only in TSP, which is a reasonable bias as we mentioned in Section 3. The hyperparameter setting is as follows:

| | TSP | CVRP |
|---|---|---|
| $\alpha$ | 0.1 | 0.2 |
| $\beta$ | 1 | 1 |
| $K$ | 100 | 100 |
| $L$ | 2 | 2 |

Table 8: Hyperparameter Setting of Sym-NCO for POMO.

Note that the design choice of $\beta = 1$ and $K = 100$ is based on POMO's baseline setting. We just set $L = 2$, because of training efficiency. We suggest to set $L > 4$ if training resources and time-budget is sufficient;; it may increase performance further.

## C.2 Integration of $\mathcal{L}_{\text{ss}}$ and $\mathcal{L}_{\text{ps}}$

The $\mathcal{L}_{\text{ps}}$ is an extension of $\mathcal{L}_{\text{ss}}$ where it can both leverage problem symmetricity and solution symmetricity. Therefore, we can simply use $\mathcal{L}_{\text{RL-Sym}} = \mathcal{L}_{\text{ps}}$. However, some specific CO problem such as TSP has cyclic nature, which contains pre-identifiable solution symmetricity, and some method already exploit the cyclic nature. For example, the POMO [23] which is a powerful NCO model already utilizes pre-identified solution symmetricity in the training process for specific CO tasks. Therefore, we provide a general loss term $\mathcal{L}_{\text{RL-Sym}} = \mathcal{L}_{\text{ps}} + \beta\mathcal{L}_{\text{ss}}$ that can be used with POMO or similar methods for specific CO problems (TSP and CVRP). If the problem has pre-identified solution symmetricity (TSP) or has a strong cyclic nature (CVRP), we can set $\beta = 1$ to leverage solution-symmetricity more. If we do not have specific domain knowledge for the target task, then we leave $\beta = 0$, to leverage problem-symmetricity and solution-symmetricity simultaneously using only $\mathcal{L}_{\text{ps}}$.

## C.3 Multi-start Post-processing

To sample multi solutions from one solver $F_\theta$ we suggest instance augmentation method following [23]. As suggested in [23], we can generate multiple samples to ablate first node selection of decoding step by $N$. Moreover we can generate 8 samples to rotate with $0, 90, 180, 270$ degrees with reflection: $4 \times 2 = 8$. To comparison with Sym-NCO and POMO as shwon in Fig. 5 (three markers), we conduct these multi state post processing with sampling width: $1, 100, 100 \times 8$.

We, on the other hand, suggest an extended version of the instance augmentation method of [23], using random orthogonal matrix $Q$. By transforming input problem $\boldsymbol{P}$ with $Q^1, ..., Q^M$ which are

orthogonal matrices, we can sample multiple sample solutions from the $M$ symmetric problems. We used these strategies in PCTSP and OP by setting $M = 200$.

## C.4 Details of Projection Head

The projection head introduced in Section 3.2 is a simple two-layer perception with the ReLU activation function, where input/output/hidden dimensions are equals to encoder's embedding dimension (i.e. 128).

## C.5 Computing Resources and Computing Time

For training Sym-NCO, we use *NVIDIA* A100 GPU. Because POMO implementation does not support GPU parallelization, we use a single GPU for the POMO + Sym-NCO. It takes approximately two weeks to finish training POMO + Sym-NCO. For training AM + Sym-NCO, we use $4\times$ GPU, which takes approximately three days to finish training.

As mentioned in Section 5.2, we use *NVIDIA* RTX2080Ti single GPU at the test time.

# D   Additional Experiments

## D.1   Hyperparameter Tuning of $\alpha$ in CVRP

We did not tune hyperparameter much in this work because training resources were limited where Sym-NCO must be verified on several tasks and DRL-NCO architectures. Therefore, we only contain simple hyperparameter ablation for $\alpha$ in CVRP (POMO + SymNCO setting).

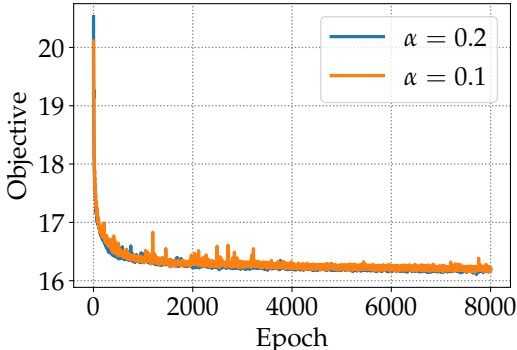

Figure 8: Alblation Study for $\alpha \in \{0.1, 0.2\}$

This validation results shows $\alpha = 0.2$ give slightly better performances than $\alpha = 0.1$, but tuning of $\alpha$ semms to be not sensitive.

## D.2 Performance Evaluation on TSPLIB

This section gives Sym-NCO performance evaluation in the TSPLIB ($N < 250$). Sym-NCO and the POMO is pre-trained model in $N = 100$ that is evaluated in Table 1. In this experiment, we conduct multi-start sampling with sample width $M = N \times 20$ where the $N$ indicates multi initial city sampling of problem size (ex., the "eil51" has $N = 51$). The 20 indicates multi-sampling using random orthogonal matrix as we introduced in Appendix C.5. As shown in the below table, our Sym-NCO outperforms POMO, having a 1.62% optimal gap, which is extremely high performance in real-world TSPLIB evaluation compared with other NCO evaluations [19].

Table 9: Performance comparison in real-world instances in TSPLIB.

| Instance | Opt. | POMO [23] | | Sym-NCO (ours) | |
|---|---|---|---|---|---|
| | | Cost | Gap | Cost | Gap |
| eil51 | 426 | 429 | 0.82% | 432 | 1.39% |
| berlin52 | 7,542 | 7,545 | 0.04% | 7,544 | 0.03% |
| st70 | 675 | 677 | 0.31% | 677 | 0.31% |
| pr76 | 108,159 | 108,681 | 0.48% | 108,388 | 0.21% |
| eil76 | 538 | 544 | 1.18% | 544 | 1.18% |
| rat99 | 1,211 | 1,270 | 4.90% | 1,261 | 4.17% |
| rd100 | 7,910 | 7,912 | 0.03% | 7,911 | 0.02% |
| KroA100 | 21,282 | 21,486 | 0.96% | 21,397 | 0.54% |
| KroB100 | 22,141 | 22,285 | 0.65% | 22,378 | 1.07% |
| KroC100 | 20,749 | 20,755 | 0.03% | 20,930 | 0.87% |
| KroD100 | 21,294 | 21,488 | 0.91% | 21,696 | 1.89% |
| KroE100 | 22,068 | 22,196 | 0.58% | 22,313 | 1.11% |
| eil101 | 629 | 641 | 1.84% | 641 | 1.84% |
| lin105 | 14,379 | 14,690 | 2.16% | 14,358 | 0.54% |
| pr124 | 59,030 | 59,353 | 0.55% | 59,202 | 0.29% |
| bier127 | 118,282 | 125,331 | 5.96% | 122,664 | 3.70% |
| ch130 | 6,110 | 6,112 | 0.03% | 6,118 | 0.14% |
| pr136 | 96,772 | 97,481 | 0.73% | 97,579 | 0.83% |
| pr144 | 58,537 | 59,197 | 1.13% | 58,930 | 0.67% |
| kroA150 | 26,524 | 26,833 | 1.16% | 26,865 | 1.28% |
| kroB150 | 26,130 | 26,596 | 1.78% | 26,648 | 1.98% |
| pr152 | 73,682 | 74,372 | 0.94% | 75,292 | 2.18% |
| u159 | 42,080 | 42.567 | 1.16% | 42,602 | 1.24% |
| rat195 | 2,323 | 2,546 | 9.58% | 2,502 | 7.70% |
| kroA200 | 29,368 | 29,937 | 1.94% | 29,816 | 1.53% |
| ts225 | 126,643 | 131,811 | 4.08% | 127,742 | 0.87% |
| tsp225 | 3,919 | 4,149 | 5.87% | 4,126 | 5.27% |
| pr226 | 80,369 | 82,428 | 2.56% | 82,337 | 2.45% |
| Avg Gap | 0.00% | 1.87% | | **1.62%** | |

### D.3 Performance Evaluation of Transferability to Large Scale Problems

This section verifies that the pre-trained model using the Sym-NCO has powerful transferability on large-scale problems. We use the efficient-active-search (EAS) [44] as a transfer learning algorithm for large-scale TSP [3]. In transfer learning, the number of iterations is an important factor for adaptation. We set the iteration $K = 200$ as default; we provide an ablation study for few-shot learning $K \in \{1, 2, 5, 10\}$ to show Sym-NCO's few-shot adaptation capability. Note that the pre-trained model is trained on CVRP ($N = 100$).

As shown in Table 10, our method outperforms the POMO [23] in large-scale CVRP, having only a small performance gap with LKH3. Furthermore, our method increase few shot adaptation capabilities for large-scale tasks, where our model achieved better performances than POMO with $5 \times$ reduced training shot $K$. To sum up, our Sym-NCO can be positioned with an effective pretraining scheme that approximately imposes symmetricity and is further transferred to larger-scale tasks.

Table 10: Performance comparison in large scale CVRP. The performance is evaluated on ten random generated CVRP data.

|  | CVRP ($N = 500$) | | CVRP ($N = 1,000$) | |
| --- | --- | --- | --- | --- |
|  | Cost | Gap | Cost | Gap |
| LKH3 [40] | 60.37 | 0.00% | 115.74 | 0.00% |
| POMO [23] + EAS [44] | 63.30 | 4.85% | 126.56 | 9.34% |
| Ours + EAS [44] | **62.41** | **3.37%** | **121.85** | **5.92%** |

Table 11: Performance evaluation of few shot adaptation to large scale CVRP.

|  | CVRP ($N = 500$) | | | | CVRP ($N = 1,000$) | | | |
| --- | --- | --- | --- | --- | --- | --- | --- | --- |
|  | $K = 1$ | $K = 2$ | $K = 5$ | $K = 10$ | $K = 1$ | $K = 2$ | $K = 5$ | $K = 10$ |
| POMO [23] + EAS [44] | 136.91 | 116.77 | 77.57 | 69.90 | 366.61 | 311.41 | 189.26 | 162.64 |
| Sym-NCO + EAS [44] | **75.85** | **69.71** | **67.26** | **66.33** | **192.12** | **163.92** | **139.66** | **134.61** |

---

[3] All the hyperparameters are the same with https://github.com/ahottung/EAS

## D.4 Comparison with Deep Improvement Heuristic Methods

In this section, we provide performance comparison with state-of-the-art deep improvement heuristics. As shown in Table 12, our method outperformed state-of-the-art deep improvement heuristics with the fastest speed. Note that constructive heuristics (which include our method) and improvement heuristics are complementary and can support each other.

Table 12: Performance comparison with deep improvement heuristics. The $I$ indicates the number of iterations, and the $s$ indicates the number of samples per instance.

|  | TSP ($N = 100$) | | TSP ($N = 100$) | |
| --- | --- | --- | --- | --- |
|  | Cost | Gap | Cost | Gap |
| Wu et al. ($I = 5K$) [16] (I=5K) | 1.42% | 2h | 2.47% | 5h |
| DACT ($I = 1K$) [25] | 1.62% | 48s | 3.18% | 2m |
| DACT ($I = 5K$) [25] | 0.61% | 4m | 1.55% | 8m |
| Ours ($s$.100) | **0.39**% | 12s | **1.46**% | 16s |
| Ours ($s$.800) | **0.14**% | 1m | **0.90**% | 2m |

We remark that the speed evaluation of Wu et al. [16] and DACT [25] is referred to by [25] where the speed is evaluated with NVIDIA TITAN RTX. The speed of our method is evaluated with NVIDIA RTX 2080Ti.

## D.5 Comparison with Symmetric NCO models

**Previous Symmetricitcy Considered NCO methods vs. Sym-NCO.** Several studies exploited the symmetric nature of CO. Ouyang et al. [37] have a similar purpose to Sym-NCO in that both are DRL-based constructive heuristics, but they give rule-based input transformation (relative position from first visited city) to satisfy equivariance. However, our method learns to impose symmetricity approximately into the neural network with regularization loss term. We believe our approach is a more general approach to tackling symmetricity (see Table 14) because not every task can be represented as a relative position with the first visited city.

The Hudson et al. [38] is the extended work of Joshi et al. [11] where graph neural network (GNN) makes a sparse graph from a fully connected input graph, and the search method figures out the feasible solution from the sparse graph. This method is based on the supervised learning scheme that requires expert labels. Moreover, this method does not guarantee to generate feasible solutions in hard-constraint CO tasks because the pruning process of the GNN may eliminate feasible trajectory (In TSP, it may work, but in other tasks, this method must address feasibility issues). Regardless of this limitation, we view the line graph transformation suggested by Hudson et al. [38] as novel and helpful in terms of symmetricity.

Ma et al. [25] proposed a DRL-based improvement heuristic, exploiting the cyclic nature of TSP and CVRP. The purpose of Ma et al.[25], and our Sym-NCO is different: the objective of Sym-NCO is approximately imposing symmetricity nature, but the objective of Ma et al. [25] is to improve the iteration process of improvement heuristic with fined designed positional encoding for TSP and CVRP. Note that Sym-NCO (constructive method) and Ma et al. [25] (Improvement method) are complementary and can support each other. For example, pretrained constructive model can generate an initial high-quality solution, whereas an improvement method can iteratively improve solution quality.

**Experimental Comparison.**

Our Sym-NCO outperforms all the relevant related baselines as shown in Table 13. Furthermore, Table 14 shows our method covers the widest arrange of CO tasks, where it does not needs labeled data.

Table 13: Performance comparison with symmetric NCO methods

|                    | Optimal Gap | Time  | GPU resources |
|--------------------|-------------|-------|---------------|
| Ouyang et al. [37] | 2.61%       | 1.3m  | GTX1080Ti     |
| Hudson et al. [38] | 0.698%      | 28h   | Tesla P100    |
| Ma et al. [25]     | 1.62%       | 48s   | Titan RTX     |
| Ours               | **0.39**%   | **12s** | RTX 2080Ti  |

Table 14: Performance comparison with symmetric NCO methods

|                    | Learning Methods        | Verified Tasks        |
|--------------------|-------------------------|-----------------------|
| Ouyang et al. [37] | Reinforcement Learning  | TSP                   |
| Hudson et al. [38] | Supervised Learning     | TSP                   |
| Ma et al. [25]     | Reinforcement Learning  | TSP, CVRP             |
| Ours               | Reinforcement Learning  | TSP, CVRP, PCTSP, OP  |