# OpenReview forum: "Sym-NCO: Leveraging Symmetricity for Neural Combinatorial Optimization"
_NeurIPS.cc/2022/Conference — NeurIPS 2022 Accept_

### Official Review · Reviewer_oKeQ · 2022-07-07

**Rating:** 6
**Confidence:** 4
**Soundness:** 2 fair
**Presentation:** 1 poor
**Contribution:** 2 fair

**Summary:**

This paper considers the problem of training neural networks to solve NP-hard CO problems. Recent works have sought to apply ML to CO, but often fall short of outperforming the state-of-the-art handcrafted heuristics; methods which neural networks have the potential to surpass in both optimality and solving time. To address this, the authors first identify that CO problems have underlying symmetries in both their problem definition and their corresponding solutions. The authors reason that explicitly learning these symmetries will implicitly aid learning to find near-optimal solutions quickly. To this end, the authors propose a new loss function which guides the network towards finding optimal solutions whilst also learning embeddings which retain these symmetries; an achievement which the authors implicitly reason makes learning to solve such CO problems more tractable for RL. Moreover, the authors claim that their loss function formulation is agnostic to the specific CO problem considered, and that it can be applied to and improve any prior neural CO solver from the literature. The authors demonstrate their proposed loss function's efficacy on four CO problems (TSP, CVRP, PCTSP, and OP), and claim to surpass all ML baselines on all four CO problems.

**Questions:**

### Introduction & Methodology

* **'Solution symmetricity and shared features' clarification:** In the Introduction when the authors introduce the two types of CO symmetry they consider, they say: ‘Second, the solution symmetricity, which is the shared feature among solutions having identical optimal values.’ I find this sentence difficult to understand - do the authors just mean that solution symmetricity is where solutions have the same value? What is the ‘shared feature’, and why do the values necessarily need to be optimal? In Definition 2.2, the authors state that two solutions are symmetric when their total returns are equal; nothing about optimality is mentioned as far as I can tell.

* **'Pre-identified symmetricites' clarification:** In the Introduction, the authors mention that their novel learning scheme ‘imposes symmetricities by leveraging the pre-identified symmetricities’. Does ‘pre-identified symmetricities’ refer to the problem and solution symmetricities, or some pre-identification method? This jargon should be explicitly defined in my view for clarity.

* **Difficulty of solution symmetricity identification:** In the Introduction, the authors state that problem symmetricity is found in all CO problems in the form of rotational symmetricity, but that solution symmetricity cannot be identified easily as the ‘properties of the solutions are distinct for every CO problem’. What specific properties are distinct? Is solution symmetricity where two solutions result in the same return (as stated in Definition 2.2), in which case why is it not easy to evaluate whether two solutions are symmetric?

* **Overall motivation and intuition:** I think the Introduction and Discussion is missing some motivation for the overall idea. Why do the authors think it is fundamentally beneficial to account for symmetricities in the learning scheme? Are there state-of-the-art solvers which do this, or are the authors relying on some intuition that explicitly learning CO symmetricities will lead to a parameterised network more able to find solutions in fewer steps since it will have learned to find multiple policies which lead to the same near-optimal reward, and therefore have greater chance of generalisation at test time? I think a discussion of what motivated the idea and what the intuition is behind it is currently missing from the paper.

* **Inconsistent jargon:** In the Sym-NCO Methodology, is the ‘invariant representation symmetry’ just the ‘problem symmetricity’ referred to elsewhere? If so, I think it would be good to keep jargon to a minimum by consistently referring to the various phenomena being discussed by the same names. If not, then I have misunderstood what is being discussed here.

* **Solution sampling methodology:** How exactly are the $K$ and $L$ solutions sampled from the REINFORCE policy? Is this done with some stochastic exploration policy?

* **Advantage function:**
    * In equation 5, is $R(\pi(P))$ the greatest rewards attained by policy $\pi$ for problem $P$ across all $K$ samples? I.e. is $R(\pi(P))$ a subset of R({\pi^{k}\}^{K}{k=1}? If not, what is the difference between how $R(\pi(P))$ and R(\{\pi^{k}\}^{K}{k=1} are generated?

    * Could the authors please explain how the proposed advantage function means the advantage will be negative if a proposed solution has a worse optimality than the K solutions sampled? Will it not only be negative if it is worse than the mean of the $K$ sampled solutions, since the baseline is just the mean return of $K$ sampled solutions? Same for $L$ in Equation 6.

* **Unclear REINFORCE methodology and integration:** Where is $L_{inv}$ (and by extension $L_{sym}$) actually incorporated into training with REINFORCE? The policy gradient theorem defined in Equations 5 and 6 only seems to include the $L_{ss}$ and $L_{ps}$ terms, so where is $L_{inv}$? Where are the three symmetric loss functions combined? Furthermore, are Equations 5 and 6 separate, or does Equation 6 contain Equation 5 with the $L$ and $K$ summations? I would like to see how all of this is tied together, since at the moment I do not think it is clear.


### Related Work

* **Missing related work and context:** This section is missing some important work and fails to introduce some of the baselines considered in the later Experiments section (e.g. S2V-DQN) and the context around them. I do not think it necessary to add all prior works to the set of baseline comparisons, since the authors are showing that the Sym-NCO method can improve a range of different architectures and methods, but I think a discussion in the related work of a few more pieces of literature should be included to put the Sym-NCO work in context. In particular, the related work is missing a discussion of where Sym-NCO fits in the context of some of the key state-of-the-art non-GNN, GNN, supervised learning, and reinforcement learning works (e.g. Bello et al. 2016, Gu and Yang 2020, Dai et al. 2017, Abe et al. 2019, Li et al. 2018, Barrett et al. 2020 and 2022, Drori et al. 2020, Hottung et al. 2022).



### Experiments

* **Small CO instances:** At $100$ nodes, the CO problems considered are very small compared to those considered by e.g. Gu and Yang 2020 ($300$ nodes), Drori et al. 2020 ($1,000$ nodes), and Barrett et al. 2022 ($10,000$ nodes). Does Sym-NCO scale? Does the requirement to sample $L \times K$ solutions for the advantage function and to gain enough data to learn the symmetricities hinder scalability? This would be interesting information to include in the paper’s experiments and discussion.

* **Statistical significance of solver performance differences:** On $O(100)$ node problems of the size considered, many of the ML-CO methods in Table 1 seem to obtain similar costs (e.g. AM gets $7.94$, POMO and MDAM get $7.80$, and Sym-NCO gets $7.79$). Is this a statistically significant difference in the optimality gap? It would seem that on such small CO problems there is not much room for differing costs, and that the results of Sym-NCO can be easily made to overfit until the reported result is achieved.

* **Optimality gap calculation:** How were the optimality gaps in Table 1 calculated? E.g. If the optimal solution of TSP is $7.76$ and Sym-NCO finds a solution of $7.84$, does this not mean that Sym-NCO’s solution cost is $1.03$% higher than the optimal solution (the authors have recorded a gap of $0.94$%)?

* **Unclear Sym-NCO integration with existing ML solvers:** Table 1 and Fig 4 do not indicate which ML method(s) Sym-NCO was applied to. They just state Sym-NCO as a standalone method, but is Sym-NCO not applied to at least one of the other baselines to get the results in the table? At the beginning of the Experiments section, the authors state that they apply Sym-NCO ‘on top of POMO, AM, and PointerNet’, but it is not clear from Table 1 and Fig 4 which underlying method was used for Sym-NCO.

* **Missing experimental data:** In Table 1, why are PointerNet and S2V-DQN missing solving time values?

* **Negative optimality gaps:** In Table 1, having a negative 'optimality gap' does not make sense - how can a solution be found which is more optimal than the optimal solution?

* **Unclear and inconsistent results:** Table 1 and Figs 3 and 4 are not consistent in multiple regards:
    * Fig 3 shows PointerNet getting a minimum validation cost of $\approx9.50$ for TSP, but Table 1 shows PointerNet to have a cost of $8.30$. Also, none of the Sym-NCO validation curves in Fig 3 reach the $7.84$/$7.79$ TSP costs claimed in Table 1. Why?
    * Fig 4a: Sym-NCO and POMO get almost exactly the same $\approx0.79$ result, yet POMO is recorded as having achieved a cost of $0.80$ in Table 1 compared to Sym-NCO’s recorded $0.79$.
    * What are the stopping criteria to stop running each solver? In Fig 4, it seems that the algorithms were still improving their solutions when they were stopped, and it also seems as though some of the other ML baselines were on track to surpass the optimality of Sym-NCO; it is important to report if Sym-NCO converges faster but to less optimal solutions.

* **Missing sensitivity analysis to the introduced hyperparameters:** What is the sensitivity of Sym-NCO to $\alpha$, $\beta$, $L$, and $K$? In the Appendix, the authors only show results for $\alpha = \{0.1, 0.2\}$, but $\alpha, \beta \in [0, 1]$. In proposing a new training scheme which introduces additional hyperparameters, the authors should have a comprehensive study of the influence of these hyperparameters on training and validation performance across different problems, since this is relevant information for practitioners who may wish to use the work.

* **Missing analysis and discussion of Sym-NCO design choices:** Which factors determine the values of $K$ and $L$ in Sym-NCO? Presumably they change as the size and nature of the CO problem changes, since some CO problems will be difficult to sample trajectories for which sufficiently encapsulate the symmetricities (e.g. some CO problems will have a vast number of possible solutions, and only a few of these might have the same objective function value; will this influence the ability with which Sym-NCO can learn the underlying solution symmetry of the problem? Is there a requirement on how many solutions with the same/similar objective value are needed to make learning solution symmetry tractable? Can solution symmetry still be imposed if no solutions with the same objective are found?)

* **Missing analysis of Sym-NCO incurred overhead:** What is the incurred training overhead of sampling the $K \times L$ solutions needed for the Sym-NCO advantage function baseline?


### Other

* **Generality claims:** The paper makes claims (in the title, abstract, and throughout the paper) to be a general neural CO solver training scheme. However, Sym-NCO has been specifically designed for policy-gradient RL and was only applied to REINFORCE. Moreover, it only considers graph-based CO problems which are variants of TSP and, for that matter, only looks at instances which can be projected onto Euclidean space. Do other ML-CO methods consider non-Euclidean problems (e.g. Dai et al. 2017, Barrett et al. 2020 & 2022, Drori et al. 2020) where graphs trained and inferred on can have differing structures? Are the authors claiming that Sym-NCO can be generalised to ML paradigms (supervised and unsupervised learning) and CO problems (non-graph-based and/or non-Euclidean graph-based) other than those considered in the paper? Can the existence of the same symmetries considered in this paper be universally assumed to exist for all CO problems?

* **Euclidean vs. non-Euclidean problem clarification:** Was there a particular reason for only considering Euclidean problems rather than utilising the now commonplace graph neural network architectures such as those used in prior works (mentioned above) which can handle non-Euclidean inputs with varying structures? What are the limitations of only handling Euclidean CO problems in terms of applications and the state-of-the-art literature? Do the authors have to hand-pick problems to meet this Euclidean constraint?


### Miscellaneous minor issues
* Pg. 1 line 2: Introduce DRL-NCO acronym but unclear what the ‘N’ stands for (presumably ‘neural’, but should specify)

* Pg. 4 line 114: Should it not be ‘as the hidden representations of $x$ and $Q(x)$’ rather than ‘$x$ and $P(x)$’?

* Pg. 4 line 119: There seems to be unnecessary extra brackets in the $g(\cdot)$ term

* Pg. 6 line 197: You list PointerNet without saying which CO problem(s) you applied it to as you did for the other methods.

* Throughout the paper, you introduce many acronyms (e.g. S2V-DQN, AM, POMO, MDAM, etc.) without first stating what the full name of the acronyms are, which you should always give when first introducing a new acronym.

* It seems confusing to refer to the method of Nazari et al. 2018 as ‘RL’ since there are multiple other RL methods such as S2V-DQN.

* Citation [20] seems to be miss-formatted?

### References

1. Hanjun Dai, Elias B. Khalil, Yuyu Zhang, Bistra Dilkina, and Le Song. Learning combinatorial optimization algorithms over graphs. In Advances in Neural Information Processing Systems, 2017
2. Thomas Barrett, William Clements, Jakob Foerster, and Alex Lvovsky. Exploratory combinatorial optimization with reinforcement learning. In Proceedings of the AAAI Conference on Artificial Intelligence, 2020
3. Iddo Drori, Anant Kharkar, William R. Sickinger, Brandon Kates, Qiang Ma, Suwen Ge, Eden Dolev, Brenda Dietrich, David P. Williamson, and Madeleine Udell. Learning to solve combinatorial optimization problems on real-world graphs in linear time. arXiv:2006.03750, 2020
4. Andre Hottung, Yeong-Dae Kwon, and Kevin Tierney. Efficient active search for combinatorial optimization problems. International Conference on Learning Representations, 2022.
5. Thomas D. Barrett, Christopher W. F. Parsonson, and Alexandre Laterre. Learning to solve combinatorial graph partitioning problems via efficient exploration. arXiv:2205.14105, 2022
6. Irwan Bello, Hieu Pham, Quoc V Le, Mohammad Norouzi, and Samy Bengio. Neural Combinatorial Optimization with Reinforcement Learning. arXiv:1611.09940, 2016
7. Shenshen Gu and Yue Yang. A Deep Learning Algorithm for the Max-Cut Problem Based on Pointer Network Structure with Supervised Learning and Reinforcement Learning Strategies. Mathematics, 2020
8. Kenshin Abe, Zijian Xu, Issei Sato, and Masashi Sugiyama. Solving NP-Hard Problems on Graphs by Reinforcement Learning without Domain Knowledge. arXiv:1905.11623, 2019
9. Zhuwen Li, Qifeng Chen, and Vladlen Koltun. Combinatorial Optimization with Graph Convolutional Networks and Guided Tree Search. In Advances in Neural Information Processing Systems, 2018
10. MohammadReza Nazari, Afshin Oroojlooy, Lawrence Snyder, and Martin Takac. Reinforcement learning for solving the vehicle routing problem. In Advances in Neural Information Processing Systems, 2018

**Strengths And Weaknesses:**

Strong points:
* Addresses an important and significant application area of ML, namely solving NP-hard CO problems.
* Proposes a novel and original loss function which guides the network towards learning the underlying symmetries of CO problems in addition to learning to find near-optimal solutions.
* The proposed approach is seemingly easy to integrate with existing ML-CO solvers and is therefore complimentary to a broad variety of prior work.

Weak points:
* The paper/writing is unclear in multiple areas (see below).
* The $100$ node problem sizes considered are significantly smaller than those of prior works.
* The Experiments section in general has multiple shortcomings and inconsistencies (see below).
* The Related Work section is incomplete and may not sufficiently place this work in the context of the current literature.

---

> ### Author Response · Authors · 2022-08-02
> **Overview of Responses**
>
> Thank you for your valuable and specific comments. They were very constructive in improving the paper. We have uploaded the revised manuscript with the modified portions marked in blue font. We would like to kindly request the reviewer to see our revised manuscript.
>
> The reviewer largely suggested four limitations, and the answers to these are briefly summarized as follows. Specific answers to individual questions will be presented further.
>
> - **Presentation**: We revised our manuscript and add figure 1 to improve the clarity of the paper.
>
> - **Large Scale CO**: We have conducted an additional experiment to test the scalability of our method with large-scale COs (CVRP N = 500, 1000). We observe that pretrained model by Sym-NCO improves scale transferability significantly. Our method achieved 3.37% (N=500) and 5.92% (N=500) gap from LKH3; while POMO shows 4.85% and 9.24%. See appendix D.3 and detailed responses regarding the scalability tests.
>
> - **Experiments inconsistencies**: These criticisms seem to stem from the unclear description of the experimental setups and results. To resolve this, we have addressed your questions in detail with concrete experiment results
>
> - **Relative Works**: We revised the discussion section to discuss the relationship between the non-Euclidean NCO models and our Sym-NCO.

---

> > ### Author Response · Authors · 2022-08-02
> > **Response of Specific Questions**
> >
> > **Question 1: Solution symmetricity and shared features' clarification**
> >
> > We agree with your comment. We have revised it into “the solution symmetricity refers to the property that solutions have identical output values” in the revised manuscript.
> >
> > ---
> >
> > **Question 2: Pre-identified symmetricites**
> >
> > The Pre-identified Symmetricities indicate the problem of symmetricity such as rotational and reflectional invariance which provably guarantees its’ symmetricity.
> >
> > ---
> >
> > **Question 3: Difficulty of solution symmetricity identification**
> >
> > Checking the solution symmetricity for the given two solutions is easy because we can just compare their solution values. However, finding a set of solutions with the same value is difficult. This is only possible for some CO problem classes whose solution structures are well understood. For example, for TSP, we know that the traveling cost of a determined route will be the same regardless of the first starting node. Thus we can identify solution symmetricity explicitly. However, for general CO problems, identifying such solution symmetricity is not straightforward, which is why we aim to learn such solution symmetricities through learning.
> >
> > ---
> >
> > **Question 4: Overall motivation and intuition**
> >
> > Leveraging symmetricity is important to train CO models for two major reasons. Firstly, symmetricity is a strong inductive bias that can support the training process of DRL by making compact training space as shown in newly updated Figure 1. Secondly, learning symmetricity is beneficial to increasing generalization capability for unseen CO problems because symmetricity induces the invariant representation that every COP contains. See figure 1 in the revised paper.
> >
> > ---
> >
> > **Question 5: Inconsistent jargon**
> >
> > The ‘invariant representation symmetricity’ and ‘problem symmetricity’ are similar but indicate different meanings.
> >
> > The Problem symmetric indicates the relationship of problems having the same optimal solution set (Def 2.1 in the main text).
> >
> > Invariant Representation Symmetricity is the objective of $L_{inv}$ that forces encoder representations from problems in problem symmetric class to contain features in some projected space.
> >
> > ---
> >
> > **Question 6: Solution sampling methodology**
> >
> > Training policy samples K x L solutions (i.e. On-policy).
> >
> > ---
> >
> > **Question 6: [Advantage function-R(π(P)) the greatest rewards attained by policy π for problem P across all K samples?]**
> >
> > **R(π(P)) is one of the sample rewards from all K samples. Specifically,**
> >
> > $\nabla L_{ss} = E_{\pi \sim F}[(R(\pi)-b)\nabla logF]$
> >
> > $\approx \frac{1}{K}\sum_{j=1}^{K}(R(\pi^{j}) - \frac{1}{K}\sum_{k=1}^{K}R(\pi^{k}))$
> >
> > **See our revised manuscript (equation 3,4,5)**
> >
> > ---
> >
> > **Question 7: [Advantage function-Could the authors please explain how the proposed advantage function means the advantage will be negative if a proposed solution has worse optimality than the K solutions sampled?]**
> >
> > In the case of solution symmetricity, we approximate the gradient of $L_{ps}$ using K sampled solutions as
> >
> > $\nabla L_{ss} = E_{\pi \sim F}[(R(\pi)-b)\nabla logF] \approx \frac{1}{K}\sum_{j=1}^{K}(R(\pi^{j}) - \frac{1}{K}\sum_{k=1}^{K}R(\pi^{k}))$
> >
> > As you said, if the sampled solution $\pi^j$ underperforms compared to the average performance of $K$ sampled solution, the advantage becomes negative. Thus, this cost term is designed to push each sampled solution to perform better than average, until all the solutions have the same value. Thus, it optimizes policy (i.e., making each solution perform better) at the same time while imposing solution symmetricity. Note that the average of advantage terms of K sampled solutions becomes zero, which means our baseline is an unbiased estimator.
> >
> > Similarly,
> >
> > $\nabla L_{ps} = E_{Q^l \sim Q}E_{\pi \sim F}[(R(\pi)-b)\nabla logF] \approx \frac{1}{LK}\sum_{i=1}^{L}\sum_{j=1}^{K}[(R(\pi^{ i,j}) - \frac{1}{LK}\sum_{i=1}^{L}\sum_{j=1}^{K}R(\pi^{i,j}))\nabla logF]$.

---

> > > ### Author Response · Authors · 2022-08-02
> > > **Continuation**
> > >
> > >
> > > **Question 7: Unclear REINFORCE methodology and integration**
> > >
> > > The total loss of Sym-NCO is $L_{total} = L_{Sym-RL} + L_{inv} = L_{pss} + L_{ss} + L_{inv}$
> > >
> > > - $L_{inv}$ is a loss term for representation learning and thus is not related to a general RL loss term (Eq.1).
> > > - Eq. 1 denotes a general RL loss term $L$ and this loss term is expended to define $L_{pss}$ and $L_{ss}$ to introduce the problem symetricity and solution symetricity, respectively, as:
> > >
> > >     $L_{ss} = E_{\pi \sim F}[R(\pi)]$
> > >
> > >     $L_{ps} = E_{Q^l \sim Q}E_{\pi \sim F}[R(\pi)]$
> > >
> > > - Eq. 5 and 6 are computed by differentiating  $L_{pss}$ and $L_{ss}$ as:
> > >
> > >     $\nabla L_{ss} = E_{\pi \sim F}[(R(\pi)-b)\nabla logF] \approx \sum_{j=1}^{K}[(R(\pi^{j}) - \frac{1}{K}\sum_{k=1}^{K}R(\pi^{k}))\nabla logF]$
> > >
> > >     $\nabla L_{ps} = E_{Q^l \sim Q}E_{\pi \sim F}[(R(\pi)-b)\nabla logF] \approx \frac{1}{LK}\sum_{i=1}^{L}\sum_{j=1}^{K}[(R(\pi^{i,j}) - \frac{1}{LK}\sum_{i=1}^{L}\sum_{j=1}^{K}R(\pi^{i,j}))\nabla logF]$
> > >
> > >
> > > The gradient of Loss is derived as a policy gradient baseline trick and approximated with the sample mean. See our revised paper.
> > >
> > > ---
> > > **Question 8: Missing related work and context**
> > >
> > > Sym-NCO in this paper focuses on integration with euclidean NCO methods. However, research for non-euclidean NCO (graph NCO) is also a very important research flow. I also mentioned in the discussion that Sym-NCO can be further extended to graph NCO models. We revised our manuscript by adding all the graph NCO literature you mentioned in the discussion section.
> > >
> > > ---
> > >
> > > **Question 9: Small CO instances**
> > >
> > > First of all, graph CO problems such as max-cut, maximum independent set (MIS), and min-cut is a locally decomposable problem, which is easier to scalable to compare with routing problems such as TSP and CVRP. Specifically, Ahn et al. 2020 proposed “learning what to defer” which hierarchical decomposes the decision process of graph CO into smaller pieces and solves over 1,000,000 nodes problem. However, Ahn et al. mentioned TSP’s routing constraint is not locally decomposable, hard to apply their scalable method.
> > >
> > > Sym-NCO also can scale more than N=100, as shown in TSPLIB results where the maximum N=250. The below table is the result using POMO + SymNCO trained in N=100, inferencing for larger-scale problems.
> > >
> > > **Exp setting**
> > >
> > > Training: same with pre-trained model reported in Table 1
> > >
> > > Inference: sampling width = 20
> > >
> > > Table for TSPLIB
> > >
> > > |  | Opt | POMO | Gap | Ours | Gap |
> > > | --- | --- | --- | --- | --- | --- |
> > > | KroA200 | 29,368 | 29,937 | 1.94% | **29,816** | **1.53%** |
> > > | ts225 | 126,643 | 131,811 | 4.08% | **127.742** | **0.87%** |
> > > | tsp225 | 3,919 | 4,149 | 5.87% | **4,126** | **5.27%** |
> > > | pr226 | 80,369 | 82,428 | 2.56% | **82,337** | **2.45%** |
> > >
> > > Furthermore, Sym-NCO can be scaled further up using (1) pertaining in small scale problem (2) transfer pretrained model to larger scale problem  (Hottung et al. 2022). Therefore we provide additional experiments on **large-scale  CVRP (N=500,1000)**.
> > >
> > > **Transfer Learning to Large Scale Experiments**
> > >
> > > Training: Pretrained model same reported in Table 1
> > >
> > > Transfer Adaptation: We referred source code of Hottung et al. 2022 (https://github.com/ahottung/EAS)
> > >
> > > - Adaptation Method: EAS-lay
> > > - Adaptation Shot: 200
> > >
> > > |  | CVRP (N=500) | CVRP (N=1000) |
> > > | --- | --- | --- |
> > > | LKH3 | 60.37 (0.00%) | 115.74 (0.00%) |
> > > | POMO + EAS {200} | 63.30 (4.85%) | 126.56 (9.24%)  |
> > > | Ours + EAS {200} | **62.41 (3.37%)** | **121.85 (5.92%)** |
> > >
> > > **Few shot Scale Transfer Adaptation Experiment**
> > > Training: Pretrained model same reported in Table 1
> > >
> > > Transfer Adaptation: We referred source code of Hottung et al. 2022 (https://github.com/ahottung/EAS)
> > >
> > > - Adaptation Method: EAS-lay
> > > - Adaptation Shot: 1,2,5,10
> > >
> > >
> > >
> > > | **CVRP ($N=500$)** | K = 1 | K=2 | K=5 | K=10 |
> > > | --- | --- | --- | --- | --- |
> > > | POMO + EAS  | 136.91 | 116.77 | 77.59 | 69.90 |
> > > | Ours + EAS | **75.85** | **69.72** | **67.26** | **66.33** |
> > >
> > > | **CVRP ($N=1,000$)** | K = 1 | K=2 | K=5 | K=10 |
> > > | --- | --- | --- | --- | --- |
> > > | POMO + EAS  | 366.61 | 311.41 | 189.26 | 162.64 |
> > > | Ours + EAS | **192.12** | **163.92** | **139.66** | **134.61** |
> > >
> > > We view large-scale routing problem research as very important further work. Sym-NCO can be positioned to a pertaining scheme. Because Sym-NCO supports learning symmetricity which is a shared feature even for large-scale problems, pertaining with Sym-NCO will help to improve scalability further.
> > >
> > > **References**
> > >
> > > - Ahn, Sungsoo, Younggyo Seo, and Jinwoo Shin. "Learning what to defer for maximum independent sets." International Conference on Machine Learning. PMLR, 2020.
> > >
> > > - André Hottung, Yeong-Dae Kwon, and Kevin Tierney. Efficient active search for combinatorial
> > > optimization problems. arXiv preprint arXiv:2106.05126, 2021.

---

> > > > ### Author Response · Authors · 2022-08-02
> > > > **Continuation**
> > > >
> > > > **Question 10: Statistical significance of solver performance difference**
> > > >
> > > > Note that the performance of the NCO model is evaluated with the test dataset which is independently separated from the training dataset.  Therefore overfitting to training dataset does not help to increase performance in test time. Our method outperforms all neural baselines in four different tasks, where some methods were “focused” on specific tasks. Also, the test dataset has 10,000 instances, the performance value is the average of them. In TSP, the optimal value is around 7.76, reducing the gap nearby the optimum not having much room for improvement. Therefore, Sym-NCO’s performance increment over POMO (which is SOTA DRL-based neural model for TSP) has meaningful satistical results.
> > > >
> > > > Furthermore, as shown in the scalability result above (CVRP N =500,1000) our model has high transferability than the baseline model which has statistical significance in avoiding overfitting.
> > > >
> > > >
> > > > ---
> > > >
> > > > **Question 11: Optimality gap calculation**
> > > >
> > > > Optimal value of TSP (N=100) with 10,000 dataset proposed by Kool et al.,: 7.76455
> > > >
> > > > POMO + SymNCO (ours): 7.8375
> > > >
> > > > Optimal Gap: (7.8375 - 7.76455)/7.76455 * 100 = 0.94%
> > > >
> > > > **Question 12: Unclear Sym-NCO integration with existing ML solvers**
> > > >
> > > > As we mentioned in section 5.2:
> > > >
> > > > Table 1: POMO + Sym-NCO
> > > >
> > > > Tabel 2: AM + Sym-NCO
> > > >
> > > >
> > > > We revised our manuscript; see table 1,2.
> > > >
> > > > ---
> > > >
> > > > **Question 13: Missing experimental data**
> > > >
> > > > We do not directly reproduce S2V-DQN because it is far from SOTA. We referred Kool et al.
> > > >
> > > > **Question 14: Negative optimality gaps**
> > > >
> > > > In PCTSP, following Kool et al., ILS is the best solver (but not the optimal solver). Therefore, the optimality gap may be misleading. The optimality gap (as Kool et al did) indicates a gap from the current best-known solver. Therefore, in PCTSP we outperformed the best-known solver, to the best of our knowledge.
> > > >
> > > > **Question 15: Unclear and inconsistent results**
> > > > That is because we reported early steps of the training process. Full training results are below:
> > > >
> > > > |  | TSP (N=100) | CVRP (N=100) |
> > > > | --- | --- | --- |
> > > > | PointerNet | 8.60 | - |
> > > > | PointerNet + Sym NCO (ours) | **8.57** | - |
> > > > | AM  | 8.12 | 16.80 |
> > > > | AM + Sym NCO (ours) | **7.90** | **16.35** |
> > > >
> > > >
> > > > Hyperparameters:
> > > >
> > > > Batchsize = 512
> > > >
> > > > Number of Epochs: 100
> > > >
> > > > Number of Instances per Epochs: 1,280,000
> > > >
> > > > L (problem sampling for L_ps) : 10
> > > >
> > > > K (solution sampling per one problem): 1
> > > >
> > > > Inference: Greedy Rollout
> > > >
> > > > Note that results of PointerNet reported in Kool et al. (2019) and reproduced model from their source code are different (Kool et al. Proposed AM, PointerNet was just for verifying their rollout baseline scheme). In Table 1, we just followed the reported value of the paper of Kool et al. (2019). We think the actual expressive power of PointerNet is not the main point of this paper, the important fact is SymNCO also can improve NCO model proposed in 2015.
> > > >
> > > > ---
> > > > **Question 16: Fig 4a**
> > > >
> > > > POMO is already a good solver in TSP as we described in section 3.2. As shown in Fig4a, the gap becomes larger when solution symmetricity is hard to identify (TSP has a trivial solution symmetricity where initial visiting can be permuted). In TSP, there is a small gap improved by Sym-NCO.
> > > >
> > > > **Question 17: Stopping Criteria**
> > > >
> > > > Stopping criteria were set by the existing paper’s stopping rule. For comparison with POMO, we follow the POMO paper’s post-processing rule. For comparison with MDAM, we also follow MDAM’s post-processing rule.
> > > >
> > > > In greedy rollout results, which is very important to see the zero-shot capability of the neural network model, Sym-NCO clearly outperformed other ML baselines. There are several other techniques for post-processing (such as 2opt). Moreover, the graph in Figure 4 is log scaled (That is why it ”seems” surpassing which is not true at all). For example, Sym-NCO zero shot greedy rollout outperforms MDAM which has 1000 times for time budget.
> > > >
> > > > ---
> > > > **Question 18: Sensitivity Analysis**
> > > >
> > > > Our training resources are not enough to tune hyperparameters because training POMO requires more than 2 weeks per task. However, we just set alpha = 0.1, and beta is 0 or 1 for every method. If we tuned hyperparameters, performance may become even more significant.
> > > >
> > > > ---
> > > > **Question 19: Missing analysis and discussion of Sym-NCO design choice**
> > > >
> > > > Yes, there are a bunch of problems that have sufficient solution symmetricity. In these cases, we can simply set K=1 but L is large (L=10 for example) to automatically identifies solution symmetricity. If there is sufficient solution symmetricity, input rotation may help to find solution symmetricity because input rotation increases the randomness of a neural network to find a different solution. If there is not sufficient solution symmetricity, problem symmetricity will be only captured by $L_{\text{inv}}$ and $L_{\text{ps}}$. See our revised manuscript Appendix C.2.

---

> > > > > ### Author Response · Authors · 2022-08-02
> > > > > **Continuation**
> > > > >
> > > > > **Question 20: Missing analysis of Sym-NCO incurred overhead**
> > > > >
> > > > > We measure VRAM allocation overhead using NVIDIA A100 single GPU, TSP ($N=100$). Overhead of POMO + SymNCO (K=100 and L can be variable) is evaluated as:
> > > > >
> > > > > | L,K | Memory |
> > > > > | --- | --- |
> > > > > | 1,100 (POMO) | 7GB |
> > > > > | 2,100 | 12GB |
> > > > > | 4,100 | 23GB |
> > > > >
> > > > > We can conclude that $L$ is directly proportional to memory consumption.
> > > > >
> > > > > ---
> > > > >
> > > > > **Question 21: Generality claim**
> > > > >
> > > > > Current Sym-NCO has verified to targets: Euclidean CO solver trained with REINFORCE (easily extended to other on-policy schemes).
> > > > >
> > > > > $L_{inv}$ term can be extended to other learning approaches: Supervised, Unsupervised Learning based Euclidean CO solver.
> > > > >
> > > > > $L_{ss}$ term can be directly extended to other domains: Graph CO solver trained with an on-policy method.
> > > > >
> > > > > $L_{ps}$ and $L_{inv}$ can be extended to graph CO domains if the proper graph CO input-transformation rule is identified.
> > > > >
> > > > > The overall concept of Sym-NCO can be extended to non-euclidean graph-based methods when the problem of symmetricity of graph input data is identified. If some graph transformation rule is founded, Sym-NCO can be directly applied to the graph CO domain.
> > > > >
> > > > > **Question 22: Euclidean vs. non-Euclidean problem clarification**
> > > > >
> > > > > We agree the non-Euclidean problem is the ultimate goal for the neural CO domain, relatively unexplored than euclidean COPs. We acknowledge and follow up with non-euclidean NCOs. Non-euclidean NCO is important because there are several important CO applications that can not be converted in euclidean nature.
> > > > >
> > > > > The reason for setting euclidean NCO as our baseline is that we wanted to show our scheme is valid in well-explored literature, which has various benchmarks and a very high-performance baseline model and easily identified symmetricity.
> > > > >
> > > > > We agree next step for COP is a non-Euclidean routing problem and the Large Scale routing problem.  To this end, our finding that leveraging symmetricity of CO is important for generalization capability is still alive for further work on non-euclidean settings and large-scale settings and can become important resources.

---

> > > > > > ### Author Response · Authors · 2022-08-02
> > > > > > **Response to Minor comments**
> > > > > >
> > > > > > - Pg. 1 line 2: Introduce DRL-NCO acronym but unclear what the ‘N’ stands for (presumably ‘neural’, but should specify)
> > > > > >
> > > > > > N means 'neural', which we already mentioned in line 26.
> > > > > >
> > > > > >
> > > > > > - Pg. 4 line 114: Should it not be ‘as the hidden representations of $x$ and $Q(x)$ rather than $x$ and $P(x)$?
> > > > > >
> > > > > > We revised it; see our revision.
> > > > > >
> > > > > > - Pg. 4 line 119: There seems to be unnecessary extra brackets in the $g(\cdot)$ term
> > > > > >
> > > > > > We revised it; see our revision.
> > > > > >
> > > > > >
> > > > > > - Pg. 6 line 197: You list PointerNet without saying which CO problem(s) you applied it to as you did for the other methods.
> > > > > >
> > > > > > We revised it; see our revision.
> > > > > >
> > > > > > - Throughout the paper, you introduce many acronyms (e.g. S2V-DQN, AM, POMO, MDAM, etc.) without first stating what the full name of the acronyms are, which you should always give when first introducing a new acronym.
> > > > > >
> > > > > > We revised it; see our revision.
> > > > > >
> > > > > > - It seems confusing to refer to the method of Nazari et al. 2018 as ‘RL’ since there are multiple other RL methods such as S2V-DQN.
> > > > > >
> > > > > > We revised as 'RL' to 'Nazari et al.'; see our revision.
> > > > > >
> > > > > > - Citation [20] seems to be miss-formatted?
> > > > > >
> > > > > > We revised it; see our revision.

---

> > > > > > > ### Comment · Reviewer_oKeQ · 2022-08-07
> > > > > > > **Reviewer oKeQ response**
> > > > > > >
> > > > > > > I thank the authors for taking the time to carefully go through and address each of my concerns and questions. I particularly like the revised version's motivation explanations and the new Figure 1. I am happy to increase my recommendation score to accept.
> > > > > > >
> > > > > > > As a final note, I suggest that the authors carefully go through their paper and correct grammatical mistakes (I think a few have been introduced in the revised text, but it would take a long time for me to list out each sentence!). It's a good paper, but perhaps e.g. Grammarly https://www.grammarly.com/ could help with some of the sentences for the camera-ready version if accepted.

---

### Official Review · Reviewer_186o · 2022-07-09

**Rating:** 6
**Confidence:** 4
**Soundness:** 2 fair
**Presentation:** 2 fair
**Contribution:** 2 fair

**Summary:**

This work is in the area of learning to approximately solve TSP, CVPR, and the associated class of routing problems using deep reinforcement learning.

The main methodological contribution is to identify that routing problems and their solutions often contain symmetries such as **rotational symmetry** or the **cyclical nature** of the solutions. The paper proposes **new loss functions** that **softly incorporate symmetries** for the encoder-decoder architecture from AM (Kool-etal) and POMO (Kwon-etal).

Training models via the proposed loss functions (termed 'Sym-NCO') improve over the models trained via previously proposed loss functions on random routing problem instances as well as on real-world instances from TSPLib.

==========

Post rebuttal: Thank you to the authors for partially addressing my concerns. I have updated my score based on the responses. In particular, I was not fully convinced regarding statements on the (lack of) expressive power of equivariant networks, e.g. “ENN scheme provably guarantees to trained in symmetric space; more expression power is needed for CO tasks”. I apologise that I am unable to engage actively in author discussions at this time as I have fallen ill after traveling.

==========

Post author discussions: Thank you for addressing my concerns regarding the claims on expressive power and hard vs. soft invariant learning. I believe this work demonstrably improves the performance of NCO solvers by leveraging appropriate symmetries, and these empirical findings may be of interest for the broader community working on combinatorial problems beyond routing. I have updated my score with these considerations.

**Questions:**

Following the **Weaknesses** part of the review, there were several parts of the paper where the writing and presentation were unclear to me:
- Lines 1-3, Abstract, states that DRL has merits over traditional CO solvers because DRL does not need supervised data. To the best of my knowledge, traditional CO solvers (which include Concorde or LKH) also **do not** need supervised data. Could the authors clarify this statement?
- I was uncertain what the relationship of Eq. 4 and Eq. 5/6 were to Eq. 1. I understand that they are components of the overall loss L_sym in Eq. 3, however, **how is L_sym used to model Eq. 1**?
- In the results in Table 1 and 2, it appears as if Sym-NCO is a model by itself. On the other hand, the experimental setting states that Sym-NCO is applied to POMO, AM, and PointerNet. Could the authors clarify which of the **underlying models** were trained for each row of results highlighted as Sym-NCO?

In addition to the questions on writing/presentation, here are additional questions regarding **approximate vs. exact rotation invariance** in Section 6.1:
- I found the results in Section 6.1 extremely counterintuitive. The results seem to suggest that provable rotation invariance is **actually undesirable**, and the authors have tried to justify this by saying that nudging the model to be approximately rotationally invariant makes it more **'flexible'** and maintain its 'representation capability'. In my opinion, the justifications are rather hand wave-y. Could the authors be more precise in defining these terms?
- As a continuation of the previous point, in my opinion, the curves in Fig. 6 show one of two things: (A) rotational symmetry is irrelevant to the problem, so EGNNs are not useful and unable to learn the task well (as seen by their fluctuating performance); or (B) the approximately rotationally invariant models are **overfitting on the data distribution** that is being used for training and validation (both are randomly generated in the same size range). Is this how one should interpret these results, or would the authors disagree?
- Finally, could the authors provide some implementation details of the EGNN used?

Finally, I had some other minor questions or nitpicks to mention:
- Titles of Sections 3.1 and 3.2 state that the symmetries are being **'imposed'**. However, in my opinion, this may be **misleading** as the symmetries are only approximated, not imposed. E.g. the encoder model's output features are only approximately invariant to rotations (unless using models which are provably rotation invariant).
- It was unclear to me why there are no empirical comparisons to the **improvement-style NCO models**. Could the authors justify this decision?

**Limitations:**

The authors have included several technical limitations of the present work and avenues for future research. They have **not included** any sections on **potential negative social impact**, and this should at least be addressed. Is it likely that this type of research may be put in production in the logistics or transportation industry, and if so, what may be some considerations to make?

In my opinion, one major limitation of this work is the **lack of justification** or understanding of **approximate vs. exact rotation invariance** in the hidden representations of the encoder. E.g. is rotational symmetry even relevant in real world datasets beyond synthetic and random TSPs/CVPRs? As such, **real world** maps and cities have a **canonical set of coordinates** and **directions** (north, south, east, west). This is to say that, in a city where I may have many locations that I would like to navigate through, I always have an aligned or fixed set of coordinates as inputs - there doesn't seem to be a good reason to arbitrarily rotate them.

It is worth considering whether methodological developments on synthetic tasks may be useful for the corresponding real-world applications in routing.

**Strengths And Weaknesses:**

Strengths:
- **Clear motivation**: The paper does a good job at presenting ideas around how routing problems and their solutions often contain symmetries such as rotational symmetry or the cyclical nature of the solutions. The figures are very instructive.

Weaknesses:
- **Unclear writing and presentation**: There are several aspects of the paper where the writing and presentation was unclear to me. I have listed this down under the **Questions** part of my review.
- **Counterintuitive results** and **lack of justification**: The results in Section 6.1 around approximate vs. exact rotation invariance are counterintuitive as they suggest to me that rotation invariance is in fact not desirable (because when it is enforced exactly, the model is unable to perform at all). However, these findings are not sufficiently justified or understood (see **Questions** section). I may have misunderstood the results or may be missing something here.
- **Insufficient discussions on related work**: There are several recent works which as based on incorporating symmetries to improve learning for routing problems, but are not discussed in this paper, e.g. [Hudson et al.](https://arxiv.org/abs/2110.05291) obtained strong results via a **rotationally invariant GNN** via converting graphs to line graphs, [Ma et al.](https://arxiv.org/abs/2110.02544) proposed positional encodings that incorporated the **cyclical nature** of routing problems, [Ouyang et al.](https://arxiv.org/abs/2110.03595) performed **preprocessing** to make the model (approximately?) invariant to rotations. The Related Work section does not do enough to contextualise present work w.r.t. recent advances in the community, which may be doing something similar or tangential to the present approach. In the best case, it would also be good to empirically compare to these techniques.

---

> ### Author Response · Authors · 2022-08-02
> **Reponse for Weekness**
>
> Thank you for your valuable comment. It was very constructive for revising our manuscript. Particularly, we could improve the presentation of the current paper based on your detailed comments.
>
> Before answering your specific question one by one, we have summarized the major changes and the effort we made to respond to the limitations the reviewer raised.
>
> **[Insufficient discussions on related work (Symmetricity based NCO)]**
>
> **Table 1** shows that our method outperforms all baselines in the fastest time. **Table 2** shows that our method covers a wide arrange of CO tasks. We have included these results in Appendix D.5
>
>
> |  **Table 1**                                    |    TSP (N=100) Optimal Gap    |    Evaluation Time    | GPU Usage          |
> |--------------------------------------|-------------------------------|-----------------------|--------------------|
> |    Ouyang et al.   (local search)    |    2.61%                      |    1.3m               |    GTX 1080Ti      |
> |    Hudson et al.   (local search)    |    0.698%                     |    28h                |    Tesla   P100    |
> |    Ma et al. ($I=1K$)                  |    1.62%                      |    4m                 | Titan RTX          |
> |    Ours (s.100)                      |    **0.39%**                      |    **12s**                | RTX   2080Ti       |
> * $I$ indicates the number iterations
> * $s$ indicates the number of sampled solution from identical problem.
>
>
> |  **Table 2**                                    |    Learning Method   |    Verified Tasks    |
> |--------------------------------------|-------------------------------|-----------------------|
> |    Ouyang et al.   (local search)    |    RL                    |    TSP           |
> |    Hudson et al.   (local search)    |    SL                     |    TSP                |
> |    Ma et al. ($I=1K$)                  |    RL                      |   TSP,CVRP                 |
> |    Ours (s.100)                      |    RL                      |    TSP,CVRP,PCTSP,OP           |
>
> ---
>
> **[Literature Review]**
>
> We have revised section 4 to add the literature review you suggested. The following is the added paragraphs to expand the literature review in appendix D.5:
>
> “Ouyang et al. have a similar purpose with Sym-NCO, in that both are DRL-based constructive heuristics, but they give rule-based input transformation (relative position from first visited city) to satisfy equivariance. However, our method learns to impose symmetricity approximately into the neural network with regularization loss term. We believe our approach is a more general approach to tackling symmetricity (see Table 2) because not every task can be represented as a relative position with the first visited city.
>
> The Hudson et al. is the extended work of Joshi et al. where graph neural network makes sparse graph from fully connected input graph, and search method figures out the feasible solution from the sparse graph. This method is based on the supervised learning scheme that requires expert labels. Moreover, this method does not guarantee to generate feasible solutions in hard-constraint CO tasks because the pruning process GNN may eliminate feasible trajectory (In TSP, it may work, but in other tasks, this method must address feasibility issues). Regardless of this limitation, we view line graph transformation as novel and helpful in terms of symmetricity.
>
> Ma et al. proposed a DRL-based improvement heuristic, exploiting the cyclic nature of TSP and CVRP. The purpose of Ma et al. and our Sym-NCO is different: the objective of Sym-NCO is approximately imposing symmetricity nature, but the objective of Ma et al. is to improve the iteration process of improvement heuristic with fined designed positional encoding for TSP and CVRP. Note that Sym-NCO (constructive method) and Ma et al. (Improvement method) are complementary and can support each other. For example, pretrained constructive model can generate an initial high-quality solution, whereas improvement method can iterative improves solution quality.”
>
> **[Counter-intuitive Results, Unclear writing, and presentation]**. We provide detailed responses below to your questions from weakness comments.
>
>
> **References**
>
> - Wenbin Ouyang, Yisen Wang, Paul Weng, and Shaochen Han. Generalization in deep rl for tsp problems via equivariance and local search. arXiv preprint arXiv:2110.03595, 2021
> - Benjamin Hudson, Qingbiao Li, Matthew Malencia, and Amanda Prorok. Graph neural network guided local search for the traveling salesperson problem. arXiv preprint arXiv:2110.05291 2021.
> - Yining Ma, Jingwen Li, Zhiguang Cao, Wen Song, Le Zhang, Zhenghua Chen, and JingTang. Learning to iteratively solve routing problems with dual-aspect collaborative transformer. Advances in Neural Information Processing Systems, 34, 2021.

---

> > ### Author Response · Authors · 2022-08-02
> > **Responses for Specific Questions**
> >
> > **Question 1: Comment of Abstract**
> >
> > The authors agree that the sentence can mislead us in that it reminds “supervised learning”. Our intention was to say DRL is able to learn an NCO solver just by interacting with the target problem (environment) without having to rely on domain expert knowledge.
> >
> > Deep reinforcement learning (DRL)-based combinatorial optimization (CO) methods (i.e., DRL-NCO) have shown significant merit over the conventional CO solvers as DRL-NCO is capable of learning CO solvers without having to rely on domain expert knowledge.”
> >
> > ---
> >
> > **Question 2: Question of Loss Terms**
> >
> > The total loss of Sym-NCO is $L_{total} = L_{inv} +L_{ps} + L_{ss}$.
> >
> > - $L_{inv}$ is a loss term for representation learning and thus is not related to a general RL loss term (Eq.1).
> > - Eq. 1 denotes a general RL loss term $L$ and this loss term is expended to define $L_{ps}$ and $L_{ss}$ to introduce the problem symetricity and solution symetricity, respectively, as: $L_{ss} = E_{\pi \sim F}[R(\pi)]$, $L_{ps} = E_{Q^l \sim Q}E_{\pi \sim F}[R(\pi)]$
> > - Eq. 5 and 6 are computed by differentiating  $L_{ps}$ and $L_{ss}$ as:
> > $\nabla L_{ss} = E_{\pi \sim F}[(R(\pi)-b)\nabla logF] \approx \frac{1}{K}\sum_{j=1}^{K}[(R(\pi^{j}) - \frac{1}{K}\sum_{k=1}^{K}R(\pi^{k}))\nabla logF]$
> > $\nabla L_{ps} = E_{Q^l \sim Q}E_{\pi \sim F}[(R(\pi)-b)\nabla logF] \approx \frac{1}{LK}\sum_{i=1}^{L}\sum_{j=1}^{K}[(R(\pi^{ i,j}) - \frac{1}{LK}\sum_{i=1}^{L}\sum_{j=1}^{K}R(\pi^{i,j}))\nabla logF]$.
> > The gradient of Loss is derived as a policy gradient baseline trick and approximated with the sample mean. See our revised paper.
> >
> > ---
> >
> > **Question 3: Sym-NCO with other Models**
> >
> > As we mentioned in the experimental setting (section 5.2), we applied Sym-NCO as follows:
> >
> > -Table 1: POMO + SymNCO.
> > -Table 2: AM + SymNCO.
> >
> > As your suggestion, we give **PointerNet + Sym-NCO** and **AM+Sym-NCO** as follows:
> >
> > |  | TSP (N=100) | CVRP (N=100) |
> > | --- | --- | --- |
> > | PointerNet | 8.60 | - |
> > | PointerNet + Sym NCO (ours) | **8.57** | - |
> > | AM  | 8.12 | 16.80 |
> > | AM + Sym NCO (ours) | **7.90** | **16.35**|
> >
> >
> > **Hyperparameters**:
> >
> > - Batchsize = 512
> >
> > - Number of Epochs: 100
> >
> > - Number of Instances per Epochs: 1,280,000
> >
> > - L (problem sampling for L_ps) : 10
> >
> > - K (solution sampling per one problem): 1
> >
> > - Inference: Greedy Rollout
> >
> > Note that results of PointerNet reported in Kool et al. (2019) and reproduced model from their source code (https://github.com/wouterkool/attention-learn-to-route) are different (Kool et al. Proposed AM (2019), PointerNet was just for verifying their rollout baseline scheme). In Table 1, we just followed the reported value of the paper of Kool et al. (2019).
> >
> > We remark that PointerNet and POMO do not support PCTSP and OP in Table 2.

---

> > > ### Author Response · Authors · 2022-08-02
> > > **Continuation**
> > >
> > >
> > >
> > > **Question 4: Section 6.1 results are counterintuitive**
> > >
> > > Your question about EGNN and our method is a critical part of our research. Thank you very much for providing us with the opportunity to clarify this important topic. If we focus only on symmetricity among many conditions that the optimal solver must have, the results presented in this study may not be intuitive. That is because the EGNN having the desired symmetricity exactly does not perform well compared to Sym-NCO having the symmetricity approximately.
> > >
> > > We believe rotation symmetricity is a necessary condition for finding the optimum solver but not a sufficient condition (see the newly added figure 1 in the revised manuscript). To support this claim, we provide the following two arguments with evidence:
> > >
> > > - **Rotational symmetricity is necessary to improve performance**. Rotational invariance is an important property to improve the generalization capability of the model for the CO tasks. Although a test instance has never been exposed during training the solver, the solver can utilize the fact that the test instance also has rotation invariance, which is the reason why Sym-NCO has better generalization capability than exactly the same model without having rotational invariance. Tables 1, 2, and figure 6 clearly show that Sym-NCO utilizing problem symmetricity outperforms the same model without having rotation invariant learning.
> > > - **Rotational symmetricity is not sufficient to improve the performance of the solver.** Figure 6 shows that the EGNN having the desired symmetricity exactly does not perform well compared to Sym-NCO having the symmetricity approximately. This trend implies that rotational symmetricity is not a sufficient condition for the optimum solver. We believe such performance difference comes from the different representation power between EGNN and Sym-NCO. Our method can utilize existing powerful CO model such as AM and POMO, which has extremely high representation capability on CO tasks. Our method is simple to integrate with existing powerful NCO models, using the proposed regularization scheme. However, EGNN is difficult to be combined with such effective NCO solver architecture.
> > >
> > > We agree that it could be best that there are provable equivariant neural networks which is a constraint to satisfy an equivariant on several CO symmetricity and at the same time maintains the existing CO model’s representation capability. We leave these directions to further work.
> > >
> > > ---
> > >
> > > **Question 5: Continuation of the Question 4**
> > >
> > > For your continuation question, we firstly acknowledge helping analyze our results. However, we have degrees in both (A) and (B).
> > >
> > > - **(A) Rotational Invariance is indeed important.** As explained above, rotational invariance is an important property to improve the generalization capability of the model for the CO tasks. That is because the rotational invariance is a shared invariant feature that every CO problem contains; even unseen CO problems in the training time have rotational invariance properties, which makes the model easy to adapt to solve such a new test problem. Tables 1, 2, and figure 6 clearly show that Sym-NCO utilizing problem symmetricity outperforms the same model without having rotation invariant learning.
> > > - **(B) Our method avoids overfitting.** Our “regularization” scheme improves generalization capability. Firstly, the CO task is all about generalization because they must solve the unseen problem. Even if the training distribution and scales are identical to the test distribution the training data and test data have different instances, optimal values, and optimal solutions. Therefore, if one model is overfitted in the training dataset, it will perform poorly in the test dataset. Furthermore, our method is verified to improves generalization capability on the different scaled problems (see Appendix D.3).
> > >
> > > ---
> > >
> > > **Question 6: Implementation details of EGNN**
> > >
> > > We used 6 EGNN encoder layers,  where the embedding dimension is 128. The EGNN layer requires three input components: Edge, Coordinate, and Node. We use the distance matrix for all cities as the edge. We use city coordinate as the Coordinate. Lastly, we use demand and prize (where the feature f in Section 3.1) as input nodes; in TSP there is no such a demand and prize so we simply use zero vector. Note that decoder layer is same with POMO.
> > > [Note that We will upload source code after decision made]
> > >
> > > ---
> > >
> > > **Question 7: Symmetricity "imposed" can be a misleading expression.**
> > > We agree with this argument. We revised it, check the revised manuscript

---

> > > > ### Author Response · Authors · 2022-08-02
> > > > **Empirical Comparison with Improvement-Style NCO models.**
> > > >
> > > > Firstly reason that we do not present empirical comparisons with improvement-style NCO is that the improvement-style NCO is complemented by constructive-style NCO. Improvement-style NCO tries to make policies that can improve the current solution. Therefore, Improvement-style NCO can also improve solutions from Constructive-style NCO which can be a hybrid NCO method.
> > > >
> > > > Constructive-NCO has a strong benefit over improvement-NCO as we mentioned in the introduction: easy to generate feasible solutions in hard constraint tasks and extremely fast.
> > > >
> > > > However, we also provide an empirical comparison with state-of-the-art improvement NCO:
> > > >
> > > > |  | TSP (N=100) |  | CVRP (N=100) |  |
> > > > | --- | --- | --- | --- | --- |
> > > > |  | Gap | Time | Gap | Time |
> > > > | Wu et al. (I=5K) | 1.42% | 2h | 2.47% | 5h |
> > > > | DACT (I=1K) | 1.62% | 48s | 3.18% | 2m |
> > > > | DACT (I=5K) | 0.61% | 4m | 1.55% | 8m |
> > > > | Ours (s.100) | 0.39% | 12s | 1.46% | 16s |
> > > > | Ours (s.800) | **0.14%** | 1m | **0.90%** | 2m |
> > > >
> > > > See detailed experiments in revised Appendix D.4.
> > > >
> > > >
> > > > **References**
> > > >
> > > > - **[DACT]** Yining Ma, Jingwen Li, Zhiguang Cao, Wen Song, Le Zhang, Zhenghua Chen, and JingTang. Learning to iteratively solve routing problems with dual-aspect collaborative transformer. Advances in Neural Information Processing Systems, 34, 2021.
> > > > - Yaoxin Wu, Wen Song, Zhiguang Cao, Jie Zhang, and Andrew Lim. Learning improvement heuristics for solving routing problems, 2020

---

> > > > > ### Author Response · Authors · 2022-08-02
> > > > > **Social Impact and Limitation**
> > > > >
> > > > >
> > > > > **Negative Social Impact**
> > > > >
> > > > > Your claims for social impact are significantly valuable. We will put an extra paragraph in the main text after the decision is made.
> > > > >
> > > > > Design automation through NCO research affects various industries including logistics and transportation industries. From a negative perspective, this automation process can lead to unemployment in certain jobs. However, automation of logistics, transportation, and design automation can increase the efficiency of industries, reducing CO2 emissions (by reducing total tour length) and creating new industries and jobs.
> > > > >
> > > > > ---
> > > > >
> > > > > **Real World Usage of Rotational Invariance**
> > > > >
> > > > > The rotational invariance is a training feature for a neural network regardless of its' actual usage in test time. For example, self-supervised learning scheme trains models have invariance features of the actual image of the cat and 90 degrees rotated cat. In the real world, it is rare that a cat is rotated or strong-augmented. However, this self-supervised is beneficial to "learn" invariant features in neural networks and increases generalization capability. Sym-NCO also has a similar motivation. See our revised introduction (motivation subsection) and figure 1 to understand the motivation of leveraging symmetricity.

---

> ### Comment · Reviewer_186o · 2022-08-06
> **Post rebuttal**
>
> Thank you to the authors for partially addressing my concerns. I have updated my score based on the responses. In particular, I was not fully convinced regarding statements on the (lack of) expressive power of equivariant networks, e.g. “ENN scheme provably guarantees to trained in symmetric space; more expression power is needed for CO tasks”. I apologise that I am unable to engage in author discussions at this time as I have fallen ill after traveling.

---

> > ### Author Response · Authors · 2022-08-08
> > **Response for your question about expression power of the EGNN**
> >
> > Thank you for your response. We are always open to discuss any time at any issue.
> >
> > Routing-style combinatorial optimization problems, including TSP, are represented as fully connected input graphs. Attentive structures, including AM and POMO, give powerful expression because multi-head attention for coordinates can powerfully represent (fully) edge connections between coordinates. However, EGNN (which is mainly designed for sparse graphs) uses a simple multi-layer perceptron (MLP) to represent each relative coordinate. Designing an equivariant attentive structure for a fully connected graph is very challenging; we leave it for further research.

---

> > > ### Comment · Reviewer_186o · 2022-08-08
> > > **Disagree; explanations are handwavy at best, misleading at worst**
> > >
> > > My comment may sound aggressive so let me preface it by saying that I believe this is a borderline paper and  it brings ideas that are of interest to the community. However, it is my duty as a reviewer to raise the following concerns to the ACs and other reviewers. It may be possible that I am mistaken.
> > >
> > > ---
> > >
> > > I respectfully disagree with the above response and to the presentation of the expressive power of ENNs vs. approximately invariant models. I believe that the claims made are handwavy at best, and misleading at worst.
> > >
> > > Firstly, the authors have not precisely defined the terms 'expression power' or expressive power. Based on Fig.1, I am assuming that they are referring to the ability of a model to universally approximate any function to arbitrary accuracy. My understanding is based on this definition of expressive power, which is a precise and formal term.
> > >
> > > Thus, the main theoretical claim by the authors is that ENNs are not expressive enough to universally approximate rotation-invariant functions over sets (or fully connected graphs).
> > > Another theoretical claim is that combinatorial problems require 'more expressive power' than what is possible for ENNs.
> > > The authors are making (seemingly) formal statements, but they have not pointed to any references or provided any proofs to support these claims.
> > >
> > > > Attentive structures, including AM and POMO, give powerful expression because multi-head attention for coordinates can powerfully represent (fully) edge connections between coordinates. However, EGNN (which is mainly designed for sparse graphs) uses a simple multi-layer perceptron (MLP) to represent each relative coordinate.
> > >
> > > In my opinion, these claims are handwavy and lack justification. They are misleading for several reasons:
> > > 1. The E(n)-equivariant GNNs paper did in fact work with fully connected graphs for all their experiments. This can be verified from the manuscript as well as the code.
> > > 2. I do not believe there is any difference between message passing (where messages are constructed via MLPs on edges) vs. attentional (where messages are constructed via learnable scalar weights) in regards to expressive power. Adding attention to an architecture does not automatically equip it with higher expressivity. Attention may work better in practice, but I have yet to see a proof showing attentional aggregation being provably more powerful than message passing.
> > > 3. There is some work showing that the model from the E(n)-equivariant GNNs paper is a universal approximator for group invariant/equivariant functions, e.g. [Appendix E](https://arxiv.org/pdf/2102.09844.pdf) of their paper, and [this work from Villar et al](https://arxiv.org/abs/2106.06610).
> > > 4. So if we assume that (a) E(n)-equivariant GNNs are universal; and (b) Theorem 2.1. from this work holds, i.e. solutions have strict rotational symmetry, then in theory, E(n)-equivariant GNNs are expressive enough to learn the solution. (BTW, if Theorem 2.1. does not strictly hold, then there is a case to be made that the possible solutions lie outside the space of functions that E(n)-equivariant GNNs can learn.)
> > >
> > > > Designing an equivariant attentive structure for a fully connected graph is very challenging; we leave it for further research.
> > >
> > > I strongly disagree, I think it is trivial to replace message passing in E(n)-equivariant GNNs with an attentional aggregation. In fact, this has been done already in a popular GitHub codebase: https://github.com/lucidrains/En-transformer. The [SE(3)-Transformers](https://arxiv.org/abs/2006.10503) paper does the same, but using higher order spherical tensors instead of cartesian vectors.
> > >
> > > ---
> > >
> > > To return to my original point, I think this paper's main contribution is regarding enforcing approximate rotational invariance (the other contribution, symmetry w.r.t. the starting city, has been proposed previously in POMO). I do not think the advantage of approximate invariance over exact invariance has been justified in a rigorous manner at all.
> > >
> > > In the revision and rebuttal, the authors have tried to justify this by making statements about the expressive power of ENNs and the expressive power needed to solve combinatorial problems. Neither of these arguments are supported by any references or formal proofs. In my opinion, the justifications are handwavy and potentially misleading.
> > >
> > > I will restate that I believe this is a borderline paper, even without any rigorous or math-y justification for approximate invariance outperforming exact invariance (it may be empirical). However, I would encourage the authors to revise their presentation.

---

> > > > ### Author Response · Authors · 2022-08-09
> > > > **Reponses**
> > > >
> > > > First of all, I would like to sincerely thank you for your valuable and constructive comments in helping the authors write the manuscript more objectively. Your comments are not offensive at all, and we are rather grateful to you for acknowledging the merits of this study. And we wish you are now fully recovered from the illness.
> > > >
> > > > **[Ambiguous use of "expressive power]**
> > > >
> > > > We agree that the term "expressive power" can be misleading if one interprets it as a level of universal approximation (UA).  In this study, we use the term "expressive power" to indicate the general performance of the solver constructed based on a specific architecture or learning method. So we would like to ensure that we did not intend to deny the UA property of an equivalent neural network (ENN). Therefore, respecting the reviewer's comment, we revised ‘expression power’ terms in our paper (see motivation, novelty, and figure 1).
> > > >
> > > > ---
> > > >
> > > > **[Contemplation about the causes of performance differences between ENN and Sym-NCO]**
> > > >
> > > > We think Sym-NCO achieved the significant performance because it can efficiently utilize effective architectures proven to be effective in RL-based CO routing fields (Kool et al., Kwon et al.). We have tried to employ EGNN-type architectures in NCO; however, the performance was unsatisfactory empirically.
> > > >
> > > > We believe the unsatisfactory performance of ENN is not because of low "expressive power". We however believe local un-decomposability of routing problems has significantly different features compared with ENN’s target benchmark. ENN works are usually verified in several geometric deep learning benchmarks including point cloud and molecule. For example, SE3 transformer and EGNN are verified on the N-body system (point cloud style data) and QM9 (molecule sparse graph). The point cloud and molecule have strong local decomposability; local clustering such as K-nearest neighborhood (KNN) processing is extremely helpful and does not degrade performances and design constraints much. For example, a molecule graph can be clearly decomposed with molecule fragments (imagine the benzene-ring attached with other molecule components); several researches studied fragment-based molecule generation and optimization (Jin et al., 2018).
> > > >
> > > > On the other hand, routing problems such as TSP are not decomposable because it has a global constraint on the Hamiltonian cycle (Ahn et al., 2020). Therefore, ENN’s technical approaches such as the KNN approximation of the SE3 transformer may not directly compete with SOTA in CO.
> > > >
> > > > However, we agree that these observations do not imply that the ENN structure is not capable of solving CO problems, but we think some delicate designing process is needed to increase performance and compete with SOTA, which may require a significant amount of additional research.
> > > >
> > > > ---
> > > >
> > > > **[Clarified Novelty of Sym-NCO]**
> > > >
> > > > We agree that the main contribution of the current paper is not on rigorously analyzing the difference between the ways to impose symmetricities: hard invariant learning (ENN) vs. soft invariant learning (Sym-NCO). Reflecting the reviewer's opinion, we exclude the argument comparing the pros and cons of the two methodologies. While mainly focusing on conveying the merits of Sym-NCO for achieving excellent performance and the simplicity of implementation, we introduce ENN as another alternative method that can reflect symmetricity and explain the difficulty of directly employing ENN for solving NCO. Although we haven't provided a mathematically rigorous analysis, we hope that the results of our Sym-NCO convey to the readers the message that we can improve the generalization performance of the learned NCO solver by exploiting the symmetricities with our proposed simple but novel approximation method. And we expect this study to lead to a discussion of different ways to reflect the symmetry inherent in many combinatorial optimization problems effectively.
> > > >
> > > > ---
> > > >
> > > > **References**
> > > >
> > > > Ahn, Sungsoo, Younggyo Seo, and Jinwoo Shin. "Learning what to defer for maximum independent sets." International Conference on Machine Learning. PMLR, 2020.
> > > >
> > > > Jin, Wengong, Regina Barzilay, and Tommi Jaakkola. "Junction tree variational autoencoder for molecular graph generation." International conference on machine learning. PMLR, 2018.
> > > >
> > > > Kool, Wouter, Herke Van Hoof, and Max Welling. "Attention, learn to solve routing problems!." arXiv preprint arXiv:1803.08475 (2018).
> > > >
> > > > Kwon, Yeong-Dae, et al. "Pomo: Policy optimization with multiple optima for reinforcement learning." Advances in Neural Information Processing Systems 33 (2020): 21188-21198.

---

> > > > > ### Comment · Reviewer_186o · 2022-08-09
> > > > > **Thank you for addressing the concerns**
> > > > >
> > > > > Thank you for addressing my concerns regarding the claims on expressive power and hard vs. soft invariant learning.
> > > > >
> > > > > I find the updated Figure 1 and accompany text more convincing. I acknowledge that the previously made claims regarding the expressive power of ENNs and the required expressive power for combinatorial optimisation tasks have now been removed.
> > > > >
> > > > > I agree that this work demonstrably improves the performance of NCO solvers by leveraging appropriate symmetries, and these empirical findings may be of interest for the broader community working on combinatorial problems beyond routing. I have updated my score with these considerations.

---

### Official Review · Reviewer_5sgV · 2022-07-13

**Rating:** 7
**Confidence:** 4
**Soundness:** 3 good
**Presentation:** 2 fair
**Contribution:** 3 good

**Summary:**

Combinatorial optimization problems often have numerous symmetries. This work proposes to leverages these symmetries to improve the training of neural networks that have been proposed in other works to solve such combinatorial optimization problems.
More specifically, the paper:
  * precisely defines two types of symmetries it wants to leverage, covering both symmetries that are inherent to the formulation of these problems and symmetries that are intrinsic to the solution spaces of such problems.
  * Proposes a regularization scheme to help the neural network learn for the symmetries
  * Evaluate their approach extensively on 4 common combinatorial optimization problems.

**Questions:**

Can you detail how you came up with equations 5 and 6 ? If needs be, you can add the relevant text to the supplemental material as you did for the proof of theorem 2.1. This would really help convince me of the soundness of your approach.

What does gr refers to in table 1?

In section 5.2 you mention that you applied Sym-NCO to POMO, AM, and PointerNet. Which one did you use to gather the results in Table 1 and 2 ? It would be interesting to be able to compare the 3 versions against their corresponding baselines whenever possible.

Would it be possible to extend your your comparison with EGNN to also train the EGNN model with your loss ? I'd love to see to what extend your approach is complementary to that of EGNN, in which case you should be able to improve on the performance reached by the EGNN model by training it with your loss.




**Limitations:**

Nothing of note here.

**Strengths And Weaknesses:**

Overall the paper is well written and fairly easy to follow. However, it's unclear how the authors came up with equation 5 and 6. It looks like the authors don't need to mathematically define Lss and Lps (which is why they never do so in the paper), but instead directly tweak the corresponding gradients in order to build two regularization gradients. The paper would be much easier to follow if that was clearly stated, and the process the authors went through in order to build these 2 gradient terms was spelled out.

Since I am not sure how the authors came up with equation 5 and 6, I didn't check the correctness of the maths, so at this point I cannot vouch for the theoretical soundness of the approach. That said, the experimental results look good, so I'm optimistic that the maths will check out. I am looking forward to more explanations from the authors in the rebuttal.

The various graphs in figure 6 are drawn for varying numbers of training steps. Why not use a consistent number of steps ?

Previous work have proposed to take advantage of symmetries to improve the generalization, mainly by leveraging symmetry invariant neural networks architectures, or by leveraging problem specific symmetries. As far as i know, this is the first work that proposes a solution that can be applied to any neural network, which is a significant innovation.

The evaluation shows that Sym-NCO results in better quality of results on all 4 benchmarks the approach was evaluated on. Furthermore, it gets these results at least as fast as the fastest other approach it was compared against. This is extremely significant.

---

> ### Author Response · Authors · 2022-08-02
> **Resonpse to Questions**
>
> Thank you for your valuable comments.
>
>
> **Question 0: Why traning step of figure 6 is not consistance?**
>
> The training step was consistancly reported; we present training graph of first 50,000 step. The training graph of PointerNet is presented from 25,000 step to 50,000. That is because traning curve of PointerNet is unstable when step T<25,000.
>
> Note that full training results of POMO + Sym-NCO was reported in Table 1. Also we report PointerNet + POMO and AM + POMO at the reponses of your Question 3 below.
>
> **Question 1: Equation Clarification**
>
> The total loss of Sym-NCO is $L_{total} = L_{inv} +L_{ps} + L_{ss}$.
>
> - $L_{inv}$ is a loss term for representation learning and thus is not related to a general RL loss term (Eq.1).
> - Eq. 1 denotes a general RL loss term $L$ and this loss term is expended to define $L_{pss}$ and $L_{ss}$ to introduce the problem symetricity and solution symetricity, respectively, as: $L_{ss} = E_{\pi \sim F}[R(\pi)]$, $L_{ps} = E_{Q^l \sim Q}E_{\pi \sim F}[R(\pi)]$
> - Eq. 5 and 6 are computed by differentiating  $L_{ps}$ and $L_{ss}$ as:
> $\nabla L_{ss} = E_{\pi \sim F}[(R(\pi)-b)\nabla logF] \approx \frac{1}{K} \sum_{j=1}^{K}[(R(\pi^{j}) - \frac{1}{K}\sum_{k=1}^{K}R(\pi^{k}))\nabla logF]$
> $\nabla L_{ps} = E_{Q^l \sim Q}E_{\pi \sim F}[(R(\pi)-b)\nabla logF] \approx \frac{1}{LK}\sum_{i=1}^{L}\sum_{j=1}^{K}[(R(\pi^{ i,j}) - \frac{1}{LK}\sum_{i=1}^{L}\sum_{j=1}^{K}R(\pi^{i,j}))\nabla logF]$.
> The gradient of Loss is derived as a policy gradient baseline trick and approximated with the sample mean. See our revised paper.
>
> ---
>
> **Question 2: What does 'gr' refers to in table 1?**
>
> The gr refers to greedy rollout (rollout of maximum probability trajectory). We revised 'gr' to 'greedy' in table 1 and 2.
>
> ---
>
> **Question 3: Which model is trained by Sym-NCO in Table 1 and 2**
>
> As we mentioned in the experimental setting (section 5.2), we applied Sym-NCO as follows:
>
> -Table 1: POMO + SymNCO.
> -Table 2: AM + SymNCO.
>
> As your suggestion, we give **PointerNet + Sym-NCO** and **AM+Sym-NCO** as follows:
>
> |  | TSP (N=100) | CVRP (N=100) |
> | --- | --- | --- |
> | PointerNet | 8.60 | - |
> | PointerNet + Sym NCO (ours) | **8.57** | - |
> | AM  | 8.12 | 16.80 |
> | AM + Sym NCO (ours) | **7.90** | 16.35 |
>
>
> **Hyperparameters**:
>
> - Batchsize = 512
>
> - Number of Epochs: 100
>
> - Number of Instances per Epochs: 1,280,000
>
> - L (problem sampling for L_ps) : 10
>
> - K (solution sampling per one problem): 1
>
> - Inference: Greedy Rollout
>
> Note that results of PointerNet reported in Kool et al. (2019) and reproduced model from their source code (https://github.com/wouterkool/attention-learn-to-route) are different (Kool et al. Proposed AM (2019), PointerNet was just for verifying their rollout baseline scheme). In Table 1, we just followed the reported value of the paper of Kool et al. (2019).
>
> **Question 4: EGNN + Sym-NCO**
>
> EGNN and our method are not complimentary. EGNN explicitly imposes symmetricities through neural network architecture (hard constraint equivariant learning), while Sym-NCO imposes symmetries by imposing regulation costs. The EGNN designed for a particular symmetricity will make the regularizing cost designed to impose the symmetriciy to be exactly zero. Thus, using both approaches at the same time will not boost performance.
>
> This may sound that EGNN will be a more direct and effective approach; however, identifying proper symmetricities for each NCO is not always easy. In addition, imposing explicitly symmetries often restrict the expressive power of a network, resulting the performance degradation.
>
> We believe your suggestion is meaningful because we can use both approaches to impose different types of symmetricities. In future research, we will consider developing a more effective method by hybridizing these two approaches.
>
> **References**
>
> - Wouter Kool, Herke van Hoof, and Max Welling. Attention, learn to solve routing problems! In International Conference on Learning Representations, 2019.

---

> ### Comment · Reviewer_5sgV · 2022-08-07
> **Response to authors' rebuttal**
>
>  I would like to thank the authors for taking the time to go through my concerns and questions. In particular, their explanation of how they derived equations 5 and 6 significantly increased my confidence in the soundness of their approach and in the correctness of the paper. I now believe this paper should be accepted.

---

### Author Response · Authors · 2022-08-02
**Overall Response**

To begin with, we thank every reviewer who helps improve our manuscript.

We have revised our manuscript significantly according to the reviewers’ comments. The revised portion is indicated in blue font. We kindly request the reviewers to see our revised manuscript while reading our responses below. The major updates are as follows:
- **Motivation**. To clearly explain the motivation for utilizing symmetricity found in COPs, we have included the “motivation” paragraph in the Introduction section. We also newly included Figure 1 to conceptually and intuitively explain the benefit of Sym-NCO over existing DRL-NCO methods and Equivariant Neural Network (ENN) schemes.
- **Detailed mathematical description regarding training method**. We have provided the detailed procedure for deriving the policy gradient terms of REINFORCE algorithm with specially designed bias terms to impose both problem and solution symmetricities.
- **Additional Experiment**. We have included two additional experiment results in the supplementary document: evaluation of transferability on large-scale CVRP (D.3), comparison with the DRL improvement heuristics (D.4), and comparison between other symmetric NCOs (D.5).

The current manuscript has received positive evaluations regarding its novelty and impact:
- **Reviewer 5sgV**: the proposed method is “innovative,” having extremely significant results on four benchmarks
- **Reviewer 186o**: the proposed method has clear motivation.
- **Reviewer oKeQ**: contains a novel and original loss scheme, is easy to integrate with prior methods, and targets significant application areas of ML.

The three key points that reviewers commonly point out and the authors' responses to them are as follows:
- **Unclear derivation procedure for deriving the REINFORCE gradients (**Reviewer 186o**, **Reviewer oKeQ**)**: We have provided the detailed procedure for deriving the policy gradient term of REINFORCE algorithm with specially designed bias terms to impose both problem and solution symmetricities. Please see Section 3.1 of the revised manuscript.
- **Unclear Motivation for imposing symmetricities (**Reviewer 186o**, **Reviewer oKeQ**)**: To clearly explain the motivation for utilizing symmetricity found in COPs, we have included the “motivation” paragraph in the Introduction section. We also newly included Figure 1 to conceptually and intuitively explain the benefit of Sym-NCO over existing DRL-NCO methods and Equivariant Neural Network (ENN) schemes.
- **Scalability (**Reviewer oKeQ**)**: We have evaluated the transferability of the method on a large-scale CVRP (N=500, 1000), and included these additional results in Appendix D.3. The result shows that Sym-NCO greatly improves scale transferability, significantly outperforming previous DRL-NCO model’s transferability (5 times faster adaptation in N=500). These results clearly strengthen our claim that learning symmetricity improves trainability by giving compact training space and generalization capability.


**We provide specific responses for each reviewer**.

---

### Meta-Review · Area_Chair_4v1L · 2022-09-04

**Recommendation:** Accept
**Confidence:** Less certain

**Metareview:**

All reviewers agree that the paper presents interesting results, hence I recommend acceptance. On the other hand there are several issues which need to be addressed in the final version of the paper:
1. The authors should add the experimental results listed in the responses, as these demonstrate more convincingly the significance of the results.
2. The mathematical formulation of the problem and the description of the solution is of extremely low quality (almost made me reject the paper). For example, nothing is defined in equation 1, neither the meaning nor the possible values of the different variables: What are the nodes? What values can features take? What is a solution? Going on to Section 2.1 and 2.2, it is again unclear what a solution is (not to mention a solution sequence), hence why we care about the corresponding MDP, what are the motivations in the definition of the MDP. What is a policy? What is a solution set? And so on. These must be written in a way which is understandable to a reader who is not already very familiar with the topic.

**Award:**

No

---

### Decision · Program_Chairs · 2022-09-14

Accept